

# Dynamic interactions between iron and sulfur cycles from Arctic methane seeps

Pauline Latour[1,2], Wei-Li Hong[2,3], Simone Sauer[2,3], Arunima Sen[2], William P. Gilhooly III[4], Aivo Lepland[3], and Fotios Fouskas[4]

[1]European Institute for Marine Studies, Université de Bretagne Occidentale, 29280, Plouzané, France
[2]CAGE- Centre for Arctic Gas Hydrate, Environment and Climate, The Arctic University of Norway in Tromsø (UiT), 9019, Norway
[3]Geological Survey of Norway, Trondheim, 7040, Norway
[4]Department of Earth Sciences, Indiana University-Purdue University Indianapolis, Indianapolis, 46202, USA

*Correspondence to*: Hong, W.-L. (wei-li.hong@ngu.no)

## Abstract

Bioavailable iron is an important micro-nutrient for marine phytoplankton and therefore critical to global biogeochemical cycles. Anoxic marine sediment is a significant source of Fe(II) to the ocean. Here, we investigate how the fluxes of Fe(II), both towards the sedimentary oxic layer and across the sediment-water interface, are impacted by the high concentration and flux of porewater sulfide in cold seep environments. We present new porewater data from four recently documented cold seeps around Svalbard as well as from continental shelves and fjords in northern Norway. We quantitatively investigated porewater data first by calculating the Fe(II) fluxes towards oxidized surface sediments and bottom water and, second, applied a transport-reaction model to estimate the mass balance of several key chemical species. Sedimentary sulfur speciation data from two of the sites were used to constrain Fe(II) consumption in the shallow sediments. We showed that the iron reduction zone is usually confined to the top 10 cm of the sediments from our studied sites due to high sulfate turnover and therefore high sulfide flux. Such a thin iron reduction zone allows proportionally more Fe(II) to reach the bottom water. Rapid precipitation of pyrite occurs at the base of the iron reduction zone, where the downward diffusing Fe(II) meets upward migrating hydrogen sulfide. Dissolved $H_2$ released during pyrite formation stimulates a small but significant rate of sulfate reduction in the same horizon, which results in faster production of hydrogen sulfide and a positive feedback for iron reduction in the shallow sediment. Deeper in the sediment, where sulfate is actively consumed due to anaerobic methane oxidation, no apparent formation of pyrite is observed from the available measurements and our modeling results. This is mostly due to the relatively low availability of Fe(II) as a result of slower turnover of the less active iron mineral phases. Such an observation may contradict the use of pyrite abundance to deduce the sulfate-methane-transition-zone in past sedimentary records. A series of model sensitivity tests were performed to systematically investigate how the Fe(II) dynamics is impacted by higher deposition rate of iron (oxyhydr)oxides minerals on the seafloor and intensifying methane supply. We showed that the increases in iron reduction rate, pyrite formation rate, and Fe(II) flux are expected with higher seafloor iron (oxyhydr)oxides deposition initially. However, complicated feedbacks between Fe(II)



production and sulfate reduction pose negative feedbacks to pyrite formation in the sediments. With a larger supply of methane, Fe(II) flux towards the oxic surface sediments is initially intensified by the higher production of hydrogen sulfide until such an interplay is too fast that essentially all reactive iron minerals settled on the seafloor dissolve immediately and dissolved iron is fixed through pyrite precipitation. Such an interplay between Fe(II) and sulfide determines the distribution

of animals with chemoautotrophic symbionts which rely on sulfide as their energy source.

## 1 Introduction

In the marine sediment, decomposition of organic matter triggers a series of redox reactions which turns oxidized N, Mn, Fe, and S species to their corresponding reduced components (Froelich et al., 1979). Redox reactions in the sediment column occur in the order according to the free energy gain per mole of organic carbon oxidized (Van Cappellen and Wang

1996). The rates of such turnovers depend on factors such as the quantity/quality/type of organic matter, bottom seawater properties, and bioturbation (Aller 1990; Schulz et al., 1994; Hedges and Keil 1995; Van Cappellen and Wang 1996). While these factors are governed by processes in the water column and across the sediment-water interface, reactions triggered by an ascending methane-rich fluid can also significantly affect the turnover of these compounds. For example, it has been well documented that large amounts of sulfide produced during sulfate reduction coupled to anaerobic oxidation of methane

(AOM) can effectively scavenge reduced iron in the porewater to form authigenic sulfide minerals, a process that has been commonly documented along the global continental margin (Schulz et al., 1994; Reimers et al., 1996; Niewohner et al., 1998; Riedinger et al., 2004; Raiswell and Anderson 2005; Lim et al., 2011; Fischer et al., 2012).

Even though the rapid production of hydrogen sulfide is well-documented in cold seep environments, a quantitative description of how these chemical species impact the conventional "redox ladder" in marine sediments is still lacking.

Previous studies already discuss how the stability of iron (oxyhydr)oxides may be affected by an increasing supply of sulfide:

$R_{hyFe}$: $2Fe(OH)_3 + 4H^+ + 1H_{2(g)} \rightarrow 2Fe^{2+} + 6H_2O$
$R_{py}$: $2HS^- + 1Fe^{2+} \rightarrow 1FeS_2 + 1H_{2(g)}$

It is obvious that a higher sulfide concentration will result in a faster precipitation of pyrite (*i.e.*, faster $R_{py}$) and/or other metastable iron sulfide minerals, such as mackinawite (Rickard and Luther 2007). The formation of these minerals consumes more Fe(II) and results in faster turnover of iron (oxyhydr)oxides (i.e., faster $R_{hyFe}$). If such iron turnover produces more Fe(II) than the amount that can be precipitated as iron sulfide minerals (*i.e.*, $R_{py} < R_{hyFe}$), Fe(II) will accumulate in the

porewater and can potentially return to the sedimentary oxic layer or even to the bottom water. It is therefore expected that steeper Fe(II) concentration gradients occur at the sites with larger sulfide fluxes which indicate higher Fe(II) fluxes towards the oxic layer in the sediments. On the other hand, it is also likely that precipitation rate of iron sulfide minerals exceeds the



rate of iron reduction (*i.e.*, $R_{py} > R_{hyFe}$, assuming no other Fe(II) source) resulting in exhaustion of all the Fe(II) and Fe(OH)$_3$ in the porewater and sediments. Such a scenario leads to no Fe(II) returning to the bottom water.

Here, we focus specifically on the interplay among various C, S, and Fe species to investigate how the migration of Fe(II) in the sediments, especially across the sediment-water interface, responds to different sulfate reduction rates and

pathways, either by organic matter degradation (organoclastic) or through AOM. Iron is a critical micro-nutrient for marine phytoplankton and photosynthesis at the surface of the ocean (Geider and La Roche 1994; Boyd et al., 2000). Even though iron is very abundant on the Earth's surface, only a small fraction is bioavailable due to its persistent solid form under oxidized conditions. Soluble Fe(II), one of the most bioavailable forms of iron, can only be produced from anoxic environments (e.g., anoxic marine sediments). Diffusion of Fe(II) from the anoxic sediments to the oxic bottom water can

enhance the precipitation of ferrihydrite which has been shown to facilitate the growth of diatoms (Raiswell and Canfield 2012). The process, termed the 'shelf-to-basin iron shuttle' was proposed to describe how iron produced through benthic cycling in the shelf area can be delivered to the open ocean with repeated redox cycling (Lyons and Severmann 2006). We intended to investigate how such benthic recycling in our study regions is impacted by methane-influenced biogeochemical processes.

In this paper, we report sediment and porewater geochemical data from four Arctic methane seep sites in the Barents Sea and from northern Norwegian fjord and shelf with water depths ranging from 220 to 380 meters. Samples were taken from box cores, multi-cores, and push cores operated by a remotely operated vehicle (ROV). We focused specifically on the top <40 cm sediment interval which covers the iron reduction zone and the upper part of the sulfate reduction zone. We estimated the benthic fluxes of iron towards the sedimentary oxic layer as well as the bottom water and quantitatively

described the porewater profiles with transport-reaction modeling. We compared our estimated Fe(II) fluxes with the values reported from the study sites of Werhmann et al. (2014), where little to no sign of methane influence was documented. We also performed a series of sensitivity tests with our data-calibrated model to quantitatively characterize the interplay between the various carbon, sulfur, and iron species. We correlated the results of our sensitivity tests with the porewater profiles and seafloor observations to examine how the interplay between iron and sulfur may affect the distribution of seafloor biota. Our

results demonstrate that the different sulfate reduction pathways and rates can have a great impact on the kinetics of iron-bearing minerals and thus the fluxes of bioavailable iron both towards the oxic surface sediments and bottom water.

## 2 Study area

We present new porewater data from 13 sediment cores, and sediment data from three of the cores, recovered from four locations (Storfjordrenna, Hola trough, Ullsfjorden, and Bjørnøyrenna; (Fig. 1 and Tab. 1). In contrast to our sites, the

porewater data from Werhmann et al. (2014) show none to very little sign of methane influence and therefore provide a baseline condition for the comparison. For the locations where sediment cores were studied, Storfjordrenna is one of the largest trough fans south of Svalbard (Lucchi et al., 2012). Repeated growth and retreat of grounding glaciers shaped the





bathymetry (Dowdeswell et al., 2010; Patton et al., 2015). At this location, texturally heterogeneous marine sediments overlie glacigenic diamictite. A large quantity of methane escaping from the sediments to the water column was documented recently (Mau et al., 2017; Serov et al., 2017). Small mound features were observed at the seafloor with gas hydrate recovered from very shallow sediment depths (Hong et al., 2017). Contrasting fluid seepage behaviors were observed

between the active and inactive mounds, interpreted to reflect the different life stages of the mounds (Hong et al., 2018; Sen et al., 2018)

Hola trough is part of the continental shelf offshore the Vesterålen Islands of northern Norway (Fig. 1). It is a cross-shelf trough with a width of 12km  and water depth of around 200m (Sauer et al., 2016). Active methane seepage was documented by geophysical (Chand et al., 2008) and geochemical investigations (Sauer et al., 2015; Sauer et al., 2016). The

sedimentation rate is relatively low and the organic matter in the sediments is predominately derived from  terrestrial sources (Sauer et al., 2016). High flux of methane derived from the Late Jurassic and Early Cretaceous source rock (Sauer et al., 2015) results in rapid sulfate turnover through AOM as documented by porewater profiles (Sauer et al., 2016). Ullsfjorden is a 70-km long fjord in northern Norway (Fig. 1). The maximum water depth in the fjord is 285 m. Numerous pockmarks have been discovered in this fjord whose origin may be related to gas escape (Plassen and Vorren 2003). Glaciomarine trough fill

is the main component of sediment in Ullsfjorden (Plassen and Vorren 2003). The Bjørnøyrenna area in the north central Barents Sea (Fig. 1), is an area where the ice sheet carved the seafloor during the last glaciation and exposed  Middle-Triassic bedrock (Long et al., 1998; Andreassen et al., 2017). Densely-distributed craters have recently been reported in this area with high methane concentration in the water column (Long et al., 1998; Andreassen et al., 2017). Only a thin layer of sediments above the bedrock is present on the seafloor. With the ROV-facilitated coring device, we recovered ~10-15 cm of

sediment at most.



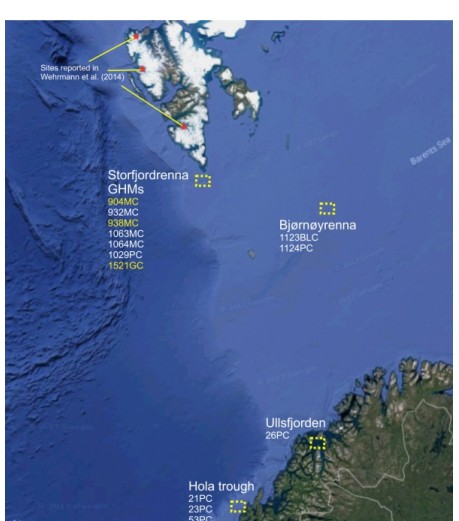

Fig.1: Map showing the four Arctic cold seeps discussed in this paper. Sites with available data for sedimentary sulfur extraction and magnetic susceptibility were labeled in yellow.

## 3 Methods

### 3.1 Coring and porewater sampling

The 13 sediment cores reported here were collected during three cruises from 2013 to 2016. Sediment cores were

5  recovered by various techniques: box corer (BC), multicorer (MC), ROV-operated push corer (PC), ROV-operated blade corer (BLC), and gravity corer (GC) onboard R/V Helmer Hanssen. Five of the multicores (CAGE15-2-904MC, CAGE15-2-932MC, CAGE15-2-938MC, CAGE16-5-1063MC, and CAGE16-5-1064MC) were recovered by using the UiT multicorer carrying a MISO-TowCam in 2015. The two push cores (CAGE16-5-1029PC and CAGE16-5-1124PC) and one blade core (CAGE16-5-1123BLC) were recovered by an ROV from the Norwegian Centre for Autonomous Marine Operations and

10  Systems (AMOS). The locations for these cores were determined based on  live seafloor images next to a gas seep in the Storfjordrenna gas hydrate mound (GHM) (1029PC) and a bacterial mat in the Bjørnøyrenna area (1123BLC) (Fig. 1). HH13-21PC, HH13-23PC, HH13-26PC and HH13-53PC were cores sub-sampled from the box cores recovered by R/V Helmer Hanssen in 2013 (Sauer et al., 2016). One gravity core, CAGE15-6-1521GC was recovered from the summit of the active GHM3 from Storfjordrenna (Hong et al., 2017; Hong et al., 2018).

15  Porewater was sampled by either 10 cm or 5 cm acid-washed rhizon samplers. Immediately after core recovery, acid-washed and MilliQ-rinsed rhizons were inserted through pre-drilled holes in the liners at 1 to 3 cm intervals, depending on the core length and anticipated redox zones. Most porewater extractions yielded >8 ml of water, with a few ranging from 1 to 5 ml, which was collected in 20 ml acid washed syringes and subsequently filtered through 0.2 μm cellulose acetate in-line



filters. Bottom water samples were carefully collected from the core top water without disturbing the sediment surface using acid-washed 20 ml syringes and treated as all other porewater samples.

Table 1:

| Site ID | Area | (A) Water Depth | (B) Bottom water Temp. | (C) $D_{bio}$ Eq(4) | (D) Fe(II)- seafloor | (E) Fe(II)- 1 cmbsf | (F) $C_{Fe-L}$ Eq(5) | (G) Log(k) Eq(7) | (H) KI Eq(6) | (I) $SO_4$ grad | (J) Fe(II) grad |
|---|---|---|---|---|---|---|---|---|---|---|---|
| | | m | °C | cm²/sec | µM | µM | mol/cm³ | (L/mole)³ | sec⁻¹ | mole/cm⁴ | mole/ cm⁴ |
| 21PC | Hola | 221 | 1 | 5.44E-08 | 0 | 5.20 | 5.20E-10 | 1.42E+01 | 3.71E-04 | 3.93E-07 | 3.72E-08 |
| 23PC | Hola | 219 | 1 | 5.44E-08 | 0 | 3.90 | 3.90E-10 | 1.42E+01 | 3.71E-04 | 1.49E-07 | 4.32E-09 |
| 53PC | Hola | 221 | 1 | 5.44E-08 | 0 | 16.50 | 1.65E-09 | 1.42E+01 | 3.71E-04 | 2.81E-08 | 1.17E-08 |
| 26PC | Ullsfjorden | 278 | 1 | 6.34E-08 | 0 | 16.20 | 1.62E-09 | 1.42E+01 | 3.71E-04 | 2.13E-07 | 1.49E-08 |
| 904MC | Storfjord-renna GHM | 377 | 0.6 | 6.34E-08 | 0.08 | 98.51 | 9.84E-09 | 1.42E+01 | 3.64E-04 | 2.80E-07 | 9.84E-08 |
| 932MC | Storfjord-renna GHM | 383 | 0.65 | 6.34E-08 | 0.14 | 3.13 | 2.99E-10 | 1.42E+01 | 3.65E-04 | 4.58E-07 | 1.03E-07 |
| 938MC | Storfjord-renna GHM | 378 | 0.6 | 6.34E-08 | 0.19 | 23.15 | 2.30E-09 | 1.42E+01 | 3.64E-04 | 7.94E-08 | 2.31E-08 |
| 1063MC | Storfjord-renna GHM | 377 | 0.6 | 6.34E-08 | 0.23 | 112.71 | 1.12E-08 | 1.42E+01 | 3.64E-04 | 3.54E-08 | 1.12E-07 |
| 1064MC | Storfjord-renna GHM | 377 | 0.6 | 6.34E-08 | 0 | 101.82 | 1.02E-08 | 1.42E+01 | 3.64E-04 | 1.97E-08 | 2.34E-08 |
| 1029PC | Storfjord-renna GHM | 377 | 0.6 | 6.34E-08 | 0.24 | 10.10 | 9.86E-10 | 1.42E+01 | 3.64E-04 | 2.72E-06 | 9.90E-09 |
| 1521GC | Storfjord-renna GHM | 386 | 0.6 | NA | NA | NA | NA | NA | NA | NA | NA |
| 1123BLC | Bjørnøyrenna | 330 | 0.6 | 6.34E-08 | 0.35 | 0.63 | 2.82E-11 | 1.42E+01 | 3.64E-04 | 2.00E-06 | 6.78E-10 |
| 1124PC | Bjørnøyrenna | 330 | 0.6 | 6.34E-08 | 0.35 | 27.37 | 2.70E-09 | 1.42E+01 | 3.64E-04 | 3.39E-08 | 4.71E-08 |

NA: Not Available



Tabel 1 (cont.)

| Site ID | Area | (K)<br>SO$_4$ flux<br>mole/<br>cm$^2$/yr | (L)<br>Fe(II) flux<br>mole/<br>cm$^2$/yr | (M)<br>Fe(II) efflux<br>mole/<br>cm$^2$/yr |
|---|---|---|---|---|
| 21PC | Hola | 2.58E-05 | 1.70E-06 | 1.75E-07 |
| 23PC | Hola | 9.80E-06 | 1.97E-07 | 1.31E-07 |
| 53PC | Hola | 1.84E-06 | 5.35E-07 | 5.54E-07 |
| 26PC | Ullsfjorden | 1.40E-05 | 6.84E-07 | 5.45E-07 |
| 904MC | Storfjord-renna GHM | 1.81E-05 | 4.43E-06 | 3.26E-06 |
| 932MC | Storfjord-renna GHM | 2.96E-05 | 4.63E-06 | 9.95E-08 |
| 938MC | Storfjord-renna GHM | 5.11E-06 | 1.04E-06 | 7.61E-07 |
| 1063MC | Storfjord-renna GHM | 2.28E-06 | 5.06E-06 | 3.73E-06 |
| 1064MC | Storfjord-renna GHM | 1.27E-06 | 1.05E-06 | 3.38E-06 |
| 1029PC | Storfjord-renna GHM | 1.75E-04 | 4.46E-07 | 3.27E-07 |
| 1521GC | Storfjord-renna GHM | NA | NA | NA |
| 1123BLC | Bjørnøyrenna | 1.29E-04 | 3.05E-08 | 9.35E-09 |
| 1124PC | Bjørnøyrenna | 2.18E-06 | 2.12E-06 | 8.96E-07 |

NA: Not Available



### 3.2 Analyses of porewater geochemistry

Alkalinity was determined with a pH-controlled titration to a pH just under 4. The pH electrode was calibrated against pH 4, 7 and 10 Metrohm Instrument buffers before and once during each cruise. HCl titrant (12M Sigma-Aldrich TraceSELECT HCl diluted to 0.012M) was prepared onboard and calibrated against both a 0.01M borax standard and local
surface seawater. The same borax standard was titrated daily during the cruise to ensure the quality of the titrant. Depending on the amount of porewater recovered, we titrated 0.1 to 1 ml of sample aliquot. About 10 ml of 0.7M KCl was added to each sample to ensure that the pH electrode was fully submerged. Titrant acid was added while constantly stirring in an open 50 ml beaker. The amount of acid and pH was manually recorded during each addition. Alkalinity was calculated from the Gran function plots, which were made by plotting Gran functions against the titrant volume. Gran function is defined as:

$$(V_0+V_t) \times 10\text{-pH}$$

where $V_0$ is the initial volume of sample and $V_t$ is the volume of titrant added. The concentration of alkalinity was then estimated from the slope and intercept of the regression line from the Gran function plot. Six to ten points were used for
regression. All titrations were done less than six hours after the syringe disconnected from the rhizons.

Concentrations of total sulfide ($\Sigma HS$) were analyzed by the 'Cline method' (Cline 1969) from the porewater samples that were fixed with 23.8 mM $Zn(OAc)_2$ onboard. Precipitates were well mixed prior to the pipetting of 50 to 200 µl of solution which were diluted to a proper concentration for the analyses. After mixing the samples with a color reagent (N,N-Dimethyl-p-phenylenediamine sulfate salt and $FeCl_3+6H_2O$ dissolved in cool 18.5% reagent grade HCl), the samples were
stored without exposure to light for 10-15 minutes to complete the reaction. They were then measured spectrophotometrically with a wave length of 670 nm. The $Na_2S$ standard was made fresh every day before analyzing samples. Thirteen standards with concentrations ranging from 0.004 to 0.25 mM were made for calibration. The method can detect $\Sigma HS$ concentration down to 4 µM; however, due to the dilution of the samples we made, the detection limit is at best 20 µM for the samples close to sediment-water interface where we expect the lowest $\Sigma HS$ concentration.

Dissolved iron (Fe(II)) was determined spectrophotometrically using a ferrospectral complex in ascorbic acid (1%) at 565 nm wavelength onboard. Calibration curves were prepared from iron sulfate standards (10 points from 0.067 to 1 mg/L iron) and run daily before the analyses. Standard and ferrospectral solutions were prepared daily with anoxic MilliQ water using acid-washed volumetric flasks. Measurements were done within an hour after the water samples were extracted. The detection limit is 0.1 µM at best.

For sulfate analyses, we used a Dionex ICS – 1100 Ion Chromatograph outfitted with an AS-DV autosampler and an IonPac AS23 column (eluent: 4.5 mM $Na_2CO_3$/0.8 mM $NaHCO_3$, flow: 1ml/min) from the lab of the Geological Survey of Norway. The relative standard deviations from repeated measurements of different laboratory standards are better than 0.5% for concentrations above 0.1 mM and better than 1.8% for concentrations above 0.02 mM.





Concentrations of ammonium were determined by a colorimetric method with a Technicon AutoAnalyzer II™ component at Oregon State University. The analytical detail is documented in the EPA Criteria "EPA 600/4-79-020 Methods for Chemical Analysis of Water and Wastes" which is available online (EPA 1983).

The isotopic composition of dissolved inorganic carbon (DIC) in pore water ($\delta^{13}$C-DIC) was determined on the $CO_2$

liberated from the water after acidification with phosphoric acid. Measurements were done with a GasBench headspace analyzer coupled to a Delta V Plus mass spectrometer (Thermo, Switzerland) at ETH Zurich and at EAWAG (The Swiss Federal Institute of Aquatic Science and Technology) using a multiflow connected to an Isoprime mass spectrometer (GV Instruments, UK). The standard deviation of the $\delta^{13}$C-DIC measurements was ±0.2‰ (1 $\sigma$). The stable carbon isotope values for DIC are reported in the conventional δ notation in permil (‰) relative to V-PDB.

Concentrations of calcium and magnesium were measured by the ICP-OES (Leeman Labs Prodigy) in the W.M. Keck Collaboratory for Plasma Spectrometry at Oregon State University in the radial viewing modes or in the lab of Geological Survey of Norway with an ICP-OES Perkin Elmer 4300 Dual View with a Cross Flow GemTip nebulizer. Samples were diluted 100 times with 1% quartz-distilled nitric acid prior to analyses. Repeated IAPSO and in-house standard were measured for every 11samples to assess the instrumental accuracy and precision. Mean concentrations and 1-sigma

uncertainties were calculated from three replicate analyses. The uncertainties are generally lower than 1 mM for magnesium and 0.1 mM for calcium.

### 3.3 Analyses of sediment geochemistry and physical properties

We quantified the relative abundance of elements in three sediment cores (904MC, 938MC, and 1521GC) through X-

Ray fluorescence (XRF) core scanning. Quantification of sulfide minerals, the acid-volatile sulfide (AVS) and chromium-reducible sulfide (CRS), in 904MC and 1521GC was performed by chemical extraction. Sediments for sulfide extraction were submerged into saturated $Zn(OAc)_2$ solution onboard and kept frozen until analyses. Upon thawing the sediments, 2-3 grams of wet sediments were weighed into the extraction vessels. The extraction of AVS was performed under room temperature with a solution of 6N HCl and 15% $SnCl_2$ (Canfield et al., 1986; Fossing and Jørgensen 1989). Hydrogen sulfide

produced during the reaction was trapped with a 3% $Zn(OAc)_2$ solution. The extractant solution was separated from the sediments through filtration onto a glass fiber filter. The CRS was then extracted subsequently from the residual sediments by boiling with a solution of 1M $CrCl_2$ and concentrated HCl (Canfield et al., 1986; Fossing and Jørgensen 1989). The hydrogen sulfide produced from both extractions was trapped by 3% $Zn(OAc)_2$ solution and kept for concentration and isotopic signature determination. The concentrations were determined following the same method described for porewater

ΣHS samples. Water content of the sediment samples was determined by measuring the weight of sediments before and after an overnight drying of the bulk/wet sediments. The magnetic susceptibility was determined from 904MC, 938MC, and 1521GC by GeoTek Multi-Sensor-Core-Logger (MSCL). The content of sulfur, aluminum, and other major/trace elements were determined by the Avaatech XRF core logger with a X-ray intensity of 120 kV and a measuring resolution of 1cm.



### 3.4 Modeling configuration

A transport-reaction model was applied using both program routines CrunchFlow and a customized MATLAB code. Such a modeling strategy has been proven successful in previous work (Hong et al., 2016; 2017). MATLAB was used to

simulate advection of solid phases (i.e., sedimentation). CrunchFlow was used to simulate diffusion and the biogeochemical reactions. The reaction network consists of four major components that target the cycling of carbon, sulfur, hydrogen, and iron in the sediments. These four components are connected through different reactions. The reaction network in the model is summarized in Fig. 2. Chemical formula of these reactions can be found in the supplementary material of Hong et al. (2017). Upon burial, labile organic matter (or particulate organic carbon, POC) is decomposed through hydrolysis which produces

low molecular weight dissolved organic carbon (LMW-DOC), which is defined as glucose ($C_6H_{12}O_6$) in the model, and other macro-nutrients, such as ammonium and phosphate (POC-hydrolysis, R1 in Fig. 2). We used the C, N, and P data from a sediment trap study in the Barents Sea (Tamelander et al., 2012) to calibrate the end member ratios of organic matter in the sediments. Glucose is then further decomposed through fermentation, which produces acetate, dissolved $H_2$, and bicarbonate (fermentation, R2 in Fig. 2). The dissolved hydrogen then induces iron reduction in the surficial sediments (HyFeR, R3 in

Fig. 2). Below the zone of iron reduction, acetate and dissolved hydrogen are consumed by sulfate reduction. Hydrogen sulfide is produced through hydrogen-induced sulfate reduction (HySR, R4 in Fig. 2) while both hydrogen sulfide and bicarbonate are produced by acetate-induced sulfate reduction (AcSR, R5 in Fig. 2). Below the zone of sulfate reduction, methane is produced through microbial methanogenesis which utilizes both acetate and hydrogen (HyME and AcME, R6 and R7 in Fig. 2). Both sulfate reduction pathways are inhibited by the presence of Fe(II) (Lovley and Phillips 1987) when

its concentration is higher than 0.4 μM based on our observations from the porewater profiles. Both methanogenesis pathways cannot outcompete sulfate reduction for substrate (Oremland and Polcin 1982) when sulfate concentrations are higher than 0.1 mM, which was also chosen based on our measured profiles. Ascending methane consumes sulfate through anaerobic methane oxidation and results in distinct geochemical interfaces in the porewater profiles known as the sulfate-methane-transition-zone (SMTZ). AOM produces hydrogen while oxidizing methane to bicarbonate across the SMTZ (R8 in

Fig. 2). This hydrogen is then harvested by sulfate reducers which stimulates rapid sulfate reduction (R4 in Fig. 2). The hydrogen sulfide produced by both AcSR and HySR diffuses upwards until it meets Fe(II) where it precipitates as pyrite (PyF, R9 in Fig. 2). We assumed a one-step formation of pyrite without considering other intermediate iron sulfide minerals. Such a setup assumes no loss of materials from dissolved species to intermediate iron sulfide minerals and to pyrite; in addition, it assumes none of the intermediate reactions are too slow to critically restrain overall pyrite formation.

Admittedly, this is a simplified setup but given that we have no data constraining the intermediate steps, it is mathematically best to assume only the overall reaction. Another source of Fe(II) below the iron reduction zone is from the dissolution of less reactive iron minerals. We chose goethite (R10 in Fig. 2) as a representation of such an iron mineral phase which is less-reactive than the iron (oxyhydr)oxide phase (Raiswell and Canfield 1998), even though the presence of goethite is not confirmed at any of the study sites. We refer to this reaction dissolution $Fe_{LR}$ (LR stands for less reactive) hereafter in the




manuscript. The rapid turnover of methane by AOM also stimulates precipitation of authigenic carbonates which consumes calcium and/or magnesium from the porewater (Ca-CP and Mg-CP, R11 and R12 in Fig. 2).

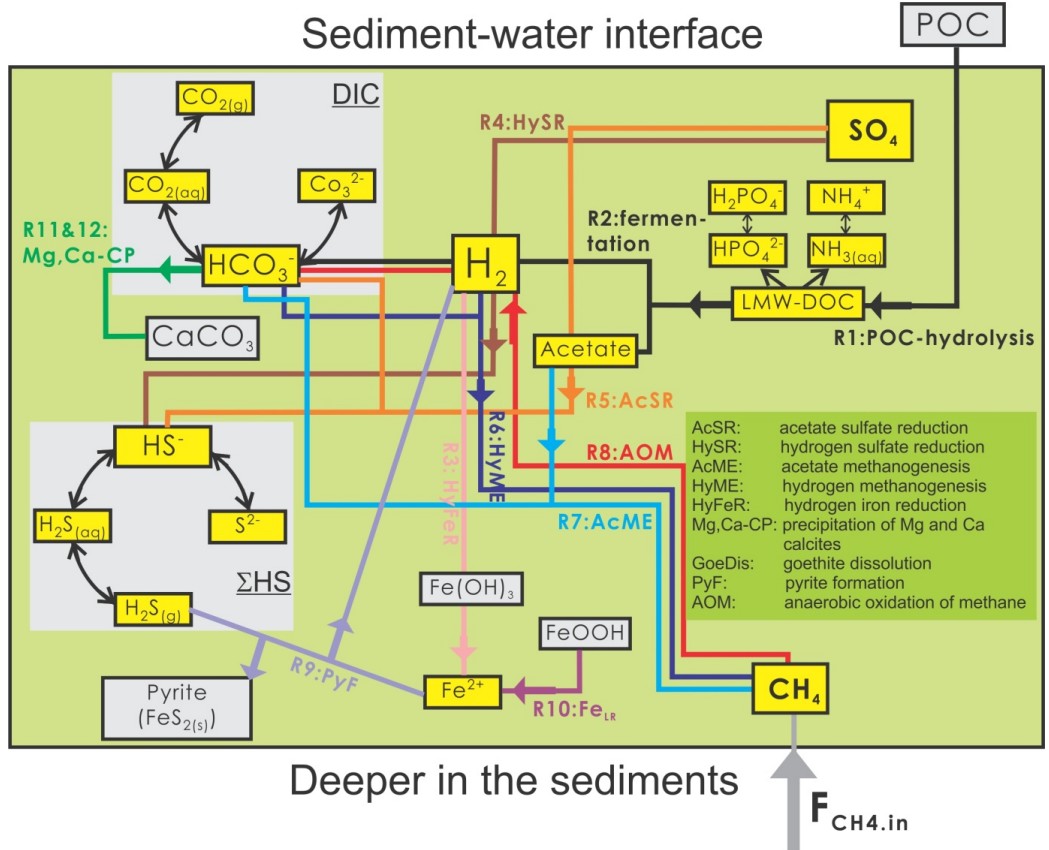

Fig. 2: The reaction network assigned in our transport-reaction model. See text and Tab. A1 in the Supplementary material for details.

For the reactions involving minerals (R1, R3, R9, R10, R11, and R12), rate expressions were formulated according to the transition state theory (Lasaga 1981) with the basic formulation of:

$$R = A_m k_m \exp[\frac{-Ea}{RT}]\prod a_i^n [1-\frac{Q}{K_{eq}}] \quad (1)$$



where $A_m$ (=1) and $k_m$ are the surface area and kinetic constant. $Ea$, $R$, and $T$ are the activation energy, ideal gas constant, and temperature. $a_i^n$ is the activity product of solutes in the reaction with their stoichiometry ($n$) as exponents. $Q/K_{eq}$, the ion activity product ($Q$) normalized to the equilibrium constant ($K_{eq}$) determines the direction of the reaction. With the exception of organic matter hydrolysis, the equilibrium constants for the other reactions were calculated from the respective Gibbs free

energy of reaction (see supplementary material in Hong et al. (2017) for the values used) assuming 25 $^{o}$C. For organic matter hydrolysis, we assumed the reaction is mostly kinetic-driven and therefore assigned an arbitrary value mainly to ensure the direction of the reaction.

For reactions involving only solutes, we formulated them as Monod-type reaction with the basic form of:

$$R = k_{max}\left(\frac{C}{C + K_{half}}\right)\left(1 - \frac{Q}{K_{eq}}\right) \quad (2)$$

where $C$ is the concentration of the focused reactant ($C_6H_{12}O_6$ for fermentation, $H_2$ for HySR and HyME, acetate for AcSR and AcME, and methane for AOM), $Q$ is the ion activity product, $K_{half}$ is the half saturation constant and $k_{max}$ is the

theoretical maximum reaction rates. $K_{eq}$, was calculated from the standard molar Gibbs free energy ($G_0^f$) of each reaction at 25 $^{o}$C.

     We used concentrations in the bottom seawater as upper boundary conditions. No exchange of solute is assumed for our lower boundary condition (i.e., Neumann boundary condition) for all chemical species. To account for the ascending methane, we set an imaginary mineral to produce methane at the deepest cell at 1.51 mbsf. The reaction rate of this

imaginary mineral therefore controls the strength of methane supply. The first 51cm of the model frame consists of 300 grids (0.17cm for each grid). We ran the simulation for 1500 years, the time required to deposit 51 cm of sediments based on the sedimentation rate (3.44E-4 cm/yr) derived from the age dates in Hong et al. (2017). We used bottom seawater composition as the initial condition. A constant porosity (0.7) is used in the modeling.

     We fitted the measured porewater profiles with the following strategy. We first fitted the ammonium profiles by

adjusting the kinetic constant of lab-hydr (R1), which determines the amount of hydrogen and acetate produced through POC hydrolysis. We then assigned the amount of iron hydroxide mineral in the surface sediments (i.e., the upper boundary condition) which decomposes through HyFeR (R3) and results in the observed concentration of Fe(II) in the porewater. HySR (R4) and AcSR (R5) are initiated when the chemical condition is suitable (i.e., a thermodynamically favorable condition) while their rates are determined by the kinetic expression we assigned. If sulfate profiles could be fitted at this

stage, we then increased the methane supply at the deepest cell which increases the AOM rate and can increase sulfate turnover through HySR (R4). Sulfide produced from both sulfate reduction pathways will consume Fe(II) through pyrite formation. In most of the cases, HyFeR (R3) alone is not sufficient to precipitate the sulfide produced. Dissolution of $Fe_{LR}$




(R10) is therefore required to produce Fe(II) in the sediment depth beneath the iron reduction zone. We assigned a dissolution-only kinetic scheme for $Fe_{LR}$; in other words, there will be no precipitation of $F_{LR}$ in the sediments even when it is thermodynamically permitted. We are aware such assumptions may not be the best. However, such a choice will not significantly affect our conclusions.

As all the above reactions can produce or consume protons, the alkalinity and pH of the pore fluid will respond to these disturbances. Alkalinity in the model is also dependent on several weak acid-base pairs including $H_2CO_3$-$HCO_3^-$-$CO_3^{2-}$, $H_2S$-$HS^-$-$S^{2-}$, $NH_4^+$-$NH_3$,and $H_3PO_4$-$H_2PO_4^-$-$HPO_4^{2-}$-$PO_4^{3-}$. We can therefore compare the model-derived total alkalinity with our titration results. For the porewater pH, even though we have no measurements to constrain, we intend to fit the model pH to a reasonable range (7-9) as reported in the literature (Schulz et al., 1994; Reimers et al., 1996; Luther et al., 1999). Dissolved

hydrogen is a central chemical species considered in our model. Despite its importance in biogeochemistry, very few concentration analyses in the porewater exist (Conrad et al., 1985; Lorenson et al., 2006). Despite the lack of constraints on hydrogen concentration, we will still discuss its mass balance as a model-based investigation.

Once the observed profiles were fitted, the model could produce reaction rates profiles of the investigated sites which we then used to calculate depth-integrated rates for various reactions. For a given chemical species, such as Fe(II), we can

estimate the flux leaving or entering the top boundary of our model by calculating the difference in depth-integrated reaction rates including all sinks and sources. When there is a net excess of an ion (i.e., higher depth-integrated rates in total sources than sinks), there must be a flux of that given ion from the sediments to the bottom water (or to the oxygenated layer, in the case of Fe(II)) to fulfill the mass balance requirement. On the other hand, when the source term is smaller than the sink term for the target ion, bottom water is then serving as a source of that ion to the sediments. With respect to Fe(II), we can

compare such model-derived flux with the flux we calculated from the measured porewater profiles (see the next paragraph for details). Note that steady-state is not a necessary assumption in our model. Any differences in the total source and sink term represent only the condition for that specific time in the system.

### 3.5 Estimation of solute fluxes from porewater profiles

We estimated the fluxes of Fe(II) and sulfate in the sediments by applying Fick's law:

$$F = \varphi . D . \frac{dc}{dx} \quad (3)$$

where $\varphi$ is the porosity (0.7), $D$ is the diffusion coefficient at different bottom water temperatures, and $\frac{dc}{dx}$ is the

concentration gradients of target chemical species (columns I and J for sulfate and Fe(II) in Tab. 1).

We consistently use 2.97E-6 cm2/sec as the diffusion coefficient for sulfate at all sites. For the diffusion coefficient of Fe(II), we considered both the molecular diffusion and bioturbation following:





$$D'_{Fe} = D_{Fe} + D_{bio} \quad (4)$$

where $D'_{Fe}$ is the $D$ in Eq. (3) for Fe(II), and $D_{Fe}$ is the the tortuosity-corrected molecular diffusion coefficient (which ranges

from 1.98E-6 to 2.01E-6 cm$^2$/sec) at different bottom water temperatures, and $D_{Bio}$ is the coefficient for bioturbation assuming the process resembles diffusion (Boudreau 1997). We estimated $D_{Bio}$ with the empirical relationship by (Middelburg et al., 1997) (see the values for different sites from column C in Tab. 1). We reported the estimated fluxes in the columns K and L of Tab. 1.

To estimate the Fe(II) flux towards the sediment-water interface (i.e., efflux), we used the equation proposed by

Boudreau and Scott (1978):

$$F_{eff} = \frac{(\varphi.(D'_{Fe}.kl)^{0.5}.C_{Fe.L})}{(\sinh[\left(\frac{kl}{D'_{Fe}}\right)^{0.5}.L])} \quad (5)$$

where $C_{Fe.L}$ is the pore water concentration of Fe(II) at the bottom of the oxic layer in the sediments (mole/cm$^3$), and $kl$ is the

first order rate constant of Fe(II) oxidation (1/sec), and $L$ is the thickness of the oxygenated layer in cm.

Determining the value of $L$ (and therefore $C_{Fe.L}$) is challenging as our sampling resolution precludes us from such estimation. We have no measurement of oxygen concentration available to determine the thickness of such oxic layers for our sites. However, based on the available data, nitrate is usually absent at the first cm below seafloor indicating a very thin nitrate reduction zone (Fig. 3) and therefore an even thinner oxic layer in the sediments. We therefore chose a value of 0.1

cm for L. For the value of $C_{Fe.L}$, we estimated the Fe(II) at 0.1 cmbsf by interpolating the measured concentrations from 0 and 1 cmbsf (values in columns D and E of Tab. 1). The resulting concentrations varied from 0.03 to 11.25 μM (Column F in Tab. 1). The highest $C_{Fe.L}$ appears at 1063MC and 1064MC from Storfjordrenna. The $kl$ value was calculated following Millero et al. (1987):

$$kl = k[O_2][OH^-]^2 \quad (6)$$

$$\log k = 21.56 - \frac{1545}{T} - 3.29I^{0,5} + 1.52I \quad (7)$$

where $k$ is the rate constant describing the overall hydrolysis equilibria of reduced iron species (Millero et al., 1987), $I$ is the ionic strength for which we used the value 0.686 as calculated by CrunchFlow with the concentrations we assigned for

bottom seawater and $T$ is the temperature for bottom water at each site (column B in Tab. 1).

We adopted the O$_2$ concentration of 320 μM based on the values reported by Anderson et al. (1988). By assuming a pH of 8.1 (Ofstad et al., unpublished data) and p$K_w$ values at the corresponding bottom water temperature at our sites (Millero





2001), we can derive [OH$^-$] in the equation. The resulting log(k) and *kl* values for all sites were included in the columns G and H from Tab. 1. The estimated Fe(II) effluxes were listed in the column M of Tab. 1.

## 4 Results

### 4.1 Porewater geochemistry

In general, we observed a very thin nitrate reduction zone for the investigated sites where nitrate concentration data was available (Fig. 3). Concentrations of Fe(II) usually increase immediately (< 10 cmbsf) below the sediment-water interface. At certain sites, Fe(II) concentrations increase constantly until the bottom of the core (932MC and 53PC in Fig. 3B) whereas the concentration decreases below a certain depth in other cores (e.g., 904MC and 21PC in Fig. 3A). Hydrogen sulfide can be detected when there is no measurable Fe(II) in the porewater, indicating active precipitation of iron sulfide

minerals. Increases of ammonium concentrations with depth at all sites indicate active decomposition of organic matter (e.g., 26PC in Fig. 3B). Based on the downcore increase in ammonium concentrations, it is clear that the rates of organic matter decomposition are widely different among sites. Decreasing sulfate concentrations with depth is expected as both organic matter decomposition and AOM consume sulfate. The decrease is however not always monotonic. At a few sites, where high methane flux is expected (e.g., cores taken from bacterial mats), sulfate concentrations decrease dramatically across small

intervals but remain >10 mM for the rest of the cores (1029PC and 1123BLC in Fig. 3C). Such profiles may indicate a non-steady-state fluid system.

Based on the porewater profiles, we can broadly classify all 12 sediment cores (except for 1521GC) into three groups. The first group is clearly influenced by methane-related diagenesis (21PC and 904MC; Fig. 3A), as suggested by rapid decrease in sulfate concentration across a small depth range. Both ΣHS and TA increase rapidly with depth. Ammonium

concentrations increase with depth but the concentrations are no higher than a few tens of μM. By assuming a C/N molar ratio of 10 and all ammonium is produced through organic matter degradation, we can estimate that only less than 1% of the TA increase is due to organic matter degradation at most sites. Given such a low contribution to the DIC pool from organic matter degradation, significant production of DIC by AOM is required to explain the high TA values, which is consistent with the low δ$^{13}$C values of DIC at some of these sites. ΣHS concentrations at these sites are generally above 1 to 2 mM

within a very confined iron reduction zone in the first few centimeters of the sediments.

In the second group of sites (23PC, 26PC, 53PC, 932MC, 938MC, 1063MC, 1064MC, and 1124PC; Figs. 3B), the influence of methane is less noticeable compared to the sites from the first group. At most of the sites in this group, sulfate turnover is slow. ΣHS concentrations are mostly below the detection limit and TA values are close to seawater concentration. Ammonium concentrations are generally below a few μM at these sites. Two of the sites in the group, 26PC and 938MC,

however, have high sulfate turnover similar to the sites in the first group as seen from their porewater sulfate profiles. The ammonium concentrations at the two sites increase rapidly to a few hundred μM at the bottom of the cores, indicating active



organic matter degradation. ΣHS concentrations at these two sites are below 2 mM with iron reduction zones of 15 to 20 cm thick.

1123BLC and 1029PC comprise the third group (Fig. 3C). The strongest methane supply among all is expected at these two locations as bacterial mats were visually confirmed when recovering the cores by ROV. The porewater profiles have

5 complicated structures which can potentially be explained by very high methane seepage activities and an increasing supply of methane (as described in Hong et al., 2017). Bubble irrigation (Haeckel et al., 2004) or bioturbation could also be the causes of these profiles. Both the concentrations of TA and ΣHS are high; especially at 1123BLC, where ΣHS concentration increases to over 10 mM at the bottom of the core. No Fe(II) is detectable at both sites as rapid pyrite formation is expected from such fast sulfide production.





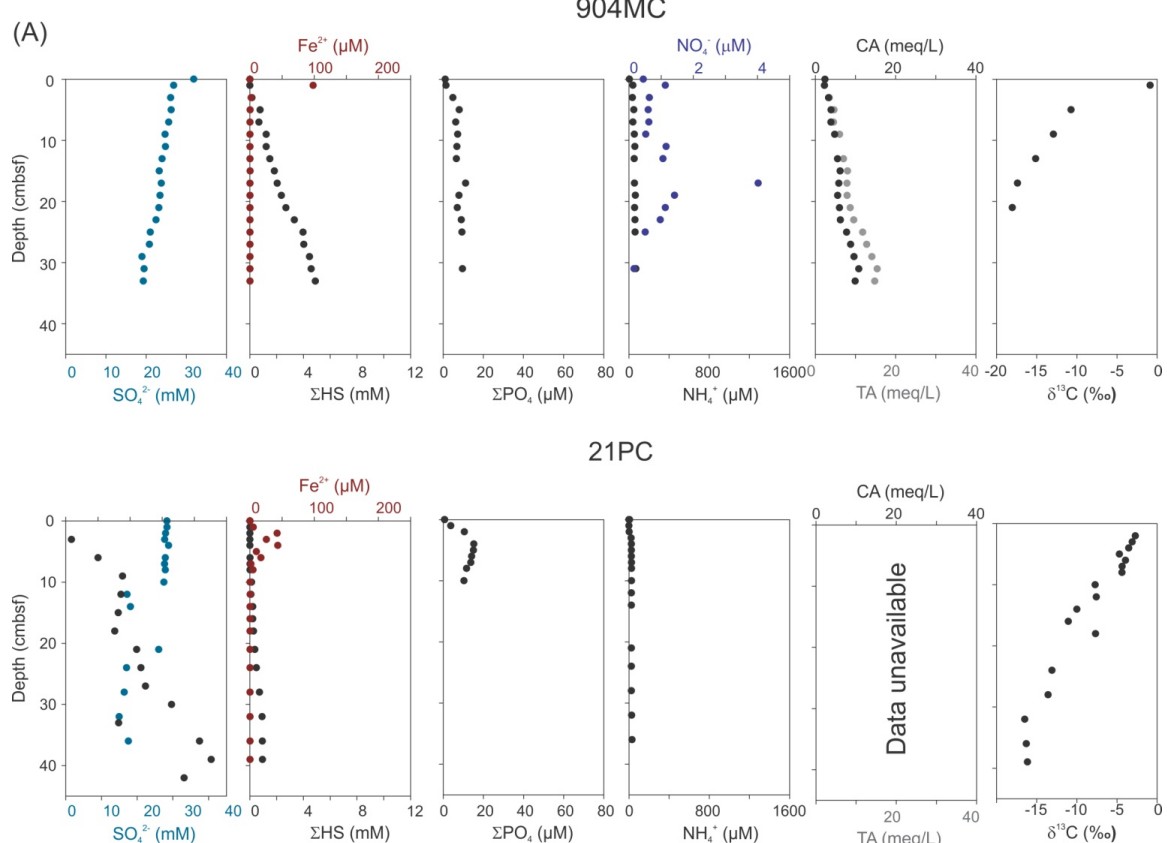

Fig. 3: Porewater profiles of the 12 investigated sediment cores (excluding 1521GC). We categorized them to: (A) sites with significant methane influence; (B) sites with little methane influence and higher contribution from organic matter degradation at some sites; (C) sites with the highest methane influence but also under a non-steady-state condition. See text for the criteria of our categorization. CA: carbonate alkalinity; TA: total alkalinity.



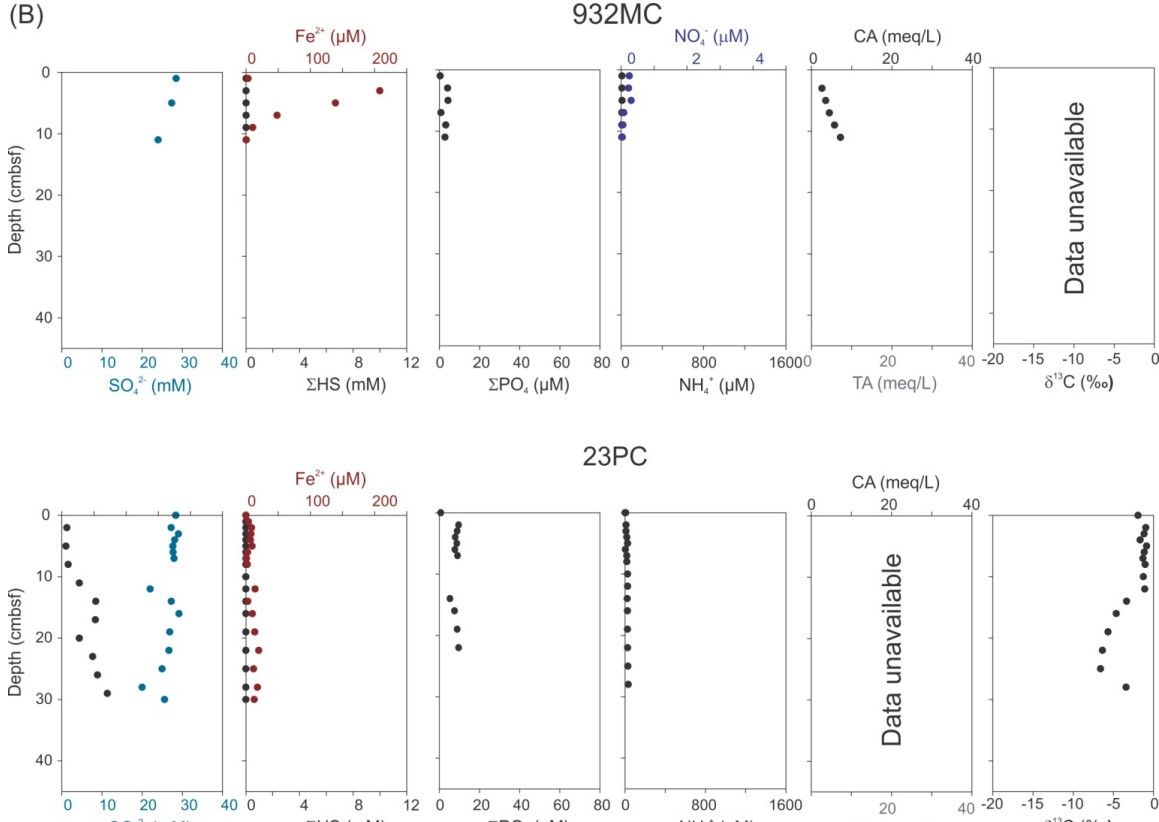



**Figure 3B (cont.)**

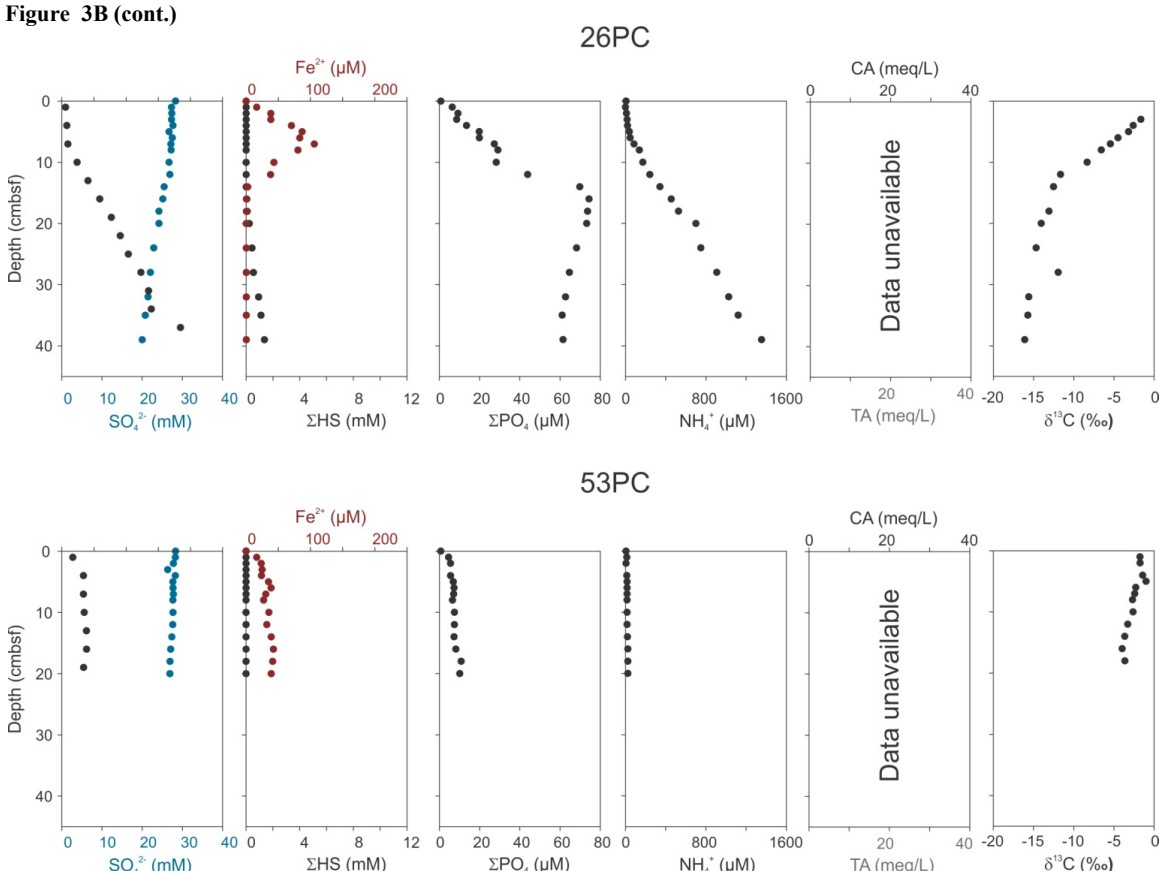



**Figure 3B (cont.)**





**Figure 3B (cont)**





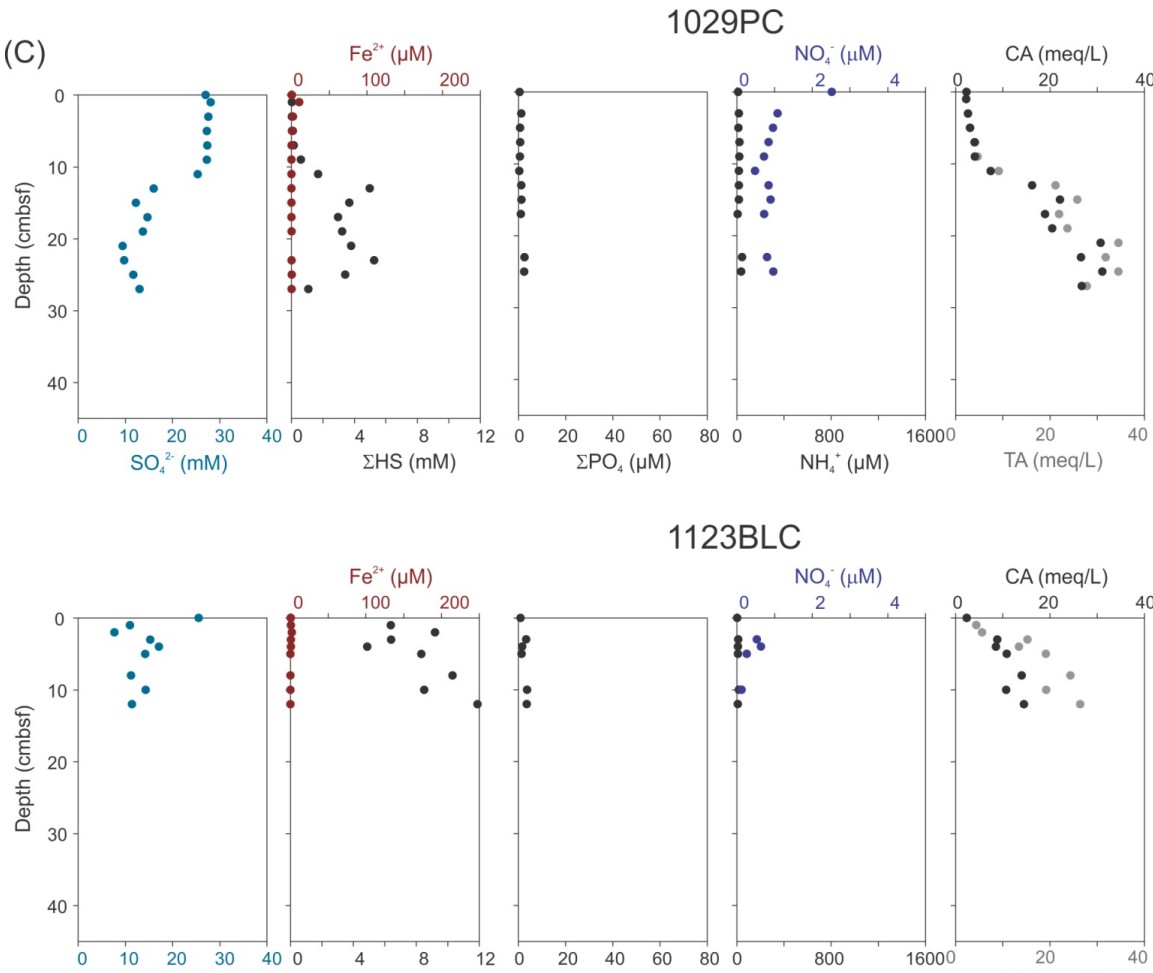



### 4.2 Sulfide speciation from the sediments

AVS is operationally defined as the sulfide minerals that can be extracted by 6M HCl at room temperature. These minerals typically include different nano-particles of iron monosulfides, such as mackinawite and greigite, and even a small fraction of pyrite (Rickard et al., 2017). CRS, on the other hand, has been shown to be comprised of mostly pyrite (Canfield

et al., 1986). Our data show variable amounts of CRS and AVS in the sediments of both 904MC and 1521GC (Fig. 4). The good agreement between the S/Al ratios and CRS abundance provides confidence that our measurements primarily track the distribution of pyrite in the sediments with minimal contribution of organic sulfur. CRS abundance increases abruptly from ca. 200 $\mu$mol S/g ($\mu$mole sulfur in gram of dry sediment) in the first few centimeters to about 1500 $\mu$mol S/g in both cores. The abundance of AVS was only measured on 1521GC where the highest abundance is observed in near-surface sediments

(154 $\mu$mol S/g at 3 cmbsf) and decreases with depth. Roughly below 30 cmbsf, AVS is mostly below the detection limit at 1521GC.

Values of magnetic susceptibility (MS) from the three investigated cores range from 3 to 18 $\times 10^{-5}$ SI unit, which are significantly lower than the values for the Holocene sediments reported around Svalbard (Jessen et al., 2010). This is expected as magnetic minerals, such as magnetite, have been shown to dissolve when exposed to high concentration of

sulfide in the porewater for considerable time (Canfield and Berner 1987; Riedinger et al., 2005; Marz et al., 2008). From 904MC, the highest MS value appears at ca. 16 cmbsf with decreasing values towards the seafloor and the bottom of the core. The highest content of CRS was detected at 11 cmbsf where MS value is very low. At 1521GC, the highest MS value was detected within the first 5cm of the sediments, where CRS content is low. Below such depth, the abundance of CRS rapidly increases with sharp decreases in MS value. The trends in MS and CRS profiles can be explained by the dissolution

of magnetic minerals and precipitation of pyrite.



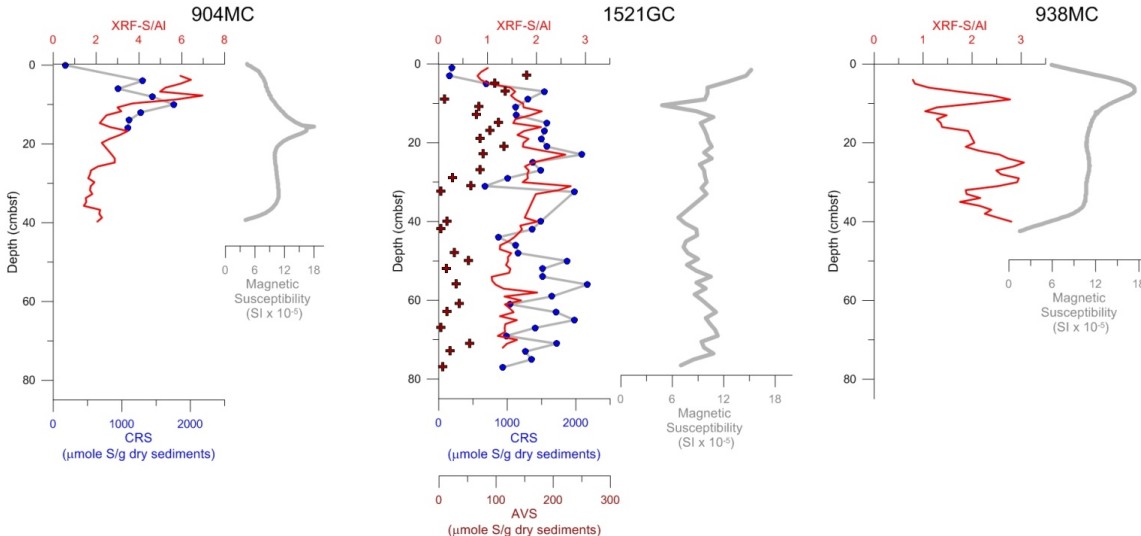

Fig. 4: The downcore variations in acid-volatile sulfur (AVS) and chromium reducible sulfur (CRS) abundance as well as the magnetic susceptibility (MS) and XRF-derived S/Al ratios at the three investigate sites from Storfjordrenna GHMs.

### 4.3 Modeling and flux calculation results

We chose porewater and solid phase data from four of the investigated sites to calibrate our model. The four sites belong to groups 1 and 2 which cover the conditions from methane-related diagenesis to organic matter fueled diagenesis. In general, we were able to fit the observed profiles (Fig. 5) and quantify the different reactions. We presented the model-derived depth-integrated rates with respect to two depth ranges, the first 51 cm of sediments (Figs. 6A-6F) and the entire length of modeling (151cm) (Figs. 6G-6L), so that we could investigate the reaction network with and without the assigned source of methane. We also presented depth-integrated rates by grouping the reactions with respect to the key chemical species associated. For example, Fe(II) is linked to three reactions: HyFeR, dissolution of $Fe_{LR}$, and pyrite formation (R3, R9, and R10 in Fig. 2; rates were shown in Figs. 6A and 6G). HyFeR is responsible for producing most of the porewater Fe(II) from three of the investigated sites. At 21PC, our model indicated that dissolution of $Fe_{LR}$ is responsible for the production of Fe(II). DIC is consumed by the precipitation of Ca/Mg authigenic carbonates and HyME (Fig. 2). Other reactions, including fermentation, AcSR, AOM, and AcME, are responsible for DIC production (Fig. 2). AOM is in general the primary source of $H_2$ among the four investigated sites whereas most of the $H_2$ at 26PC is produced through fermentation. HySR is the primary sink of $H_2$ in all four sites and it couples with either AOM or fermentation based on the substrate available. Even though we did not force the coupling between AOM and HySR in our model, it is apparent that such coupling is thermodynamically favorable when there is only limited amount of reactive organic matter. From the mass





balance of bicarbonate and $H_2$, we can infer that organic matter is the primary driver for reactions at 26PC. This is not surprising given the abundant ammonium detected in porewater (Fig. 5B) and the high organic matter content in this fjord (Sauer et al., 2016).



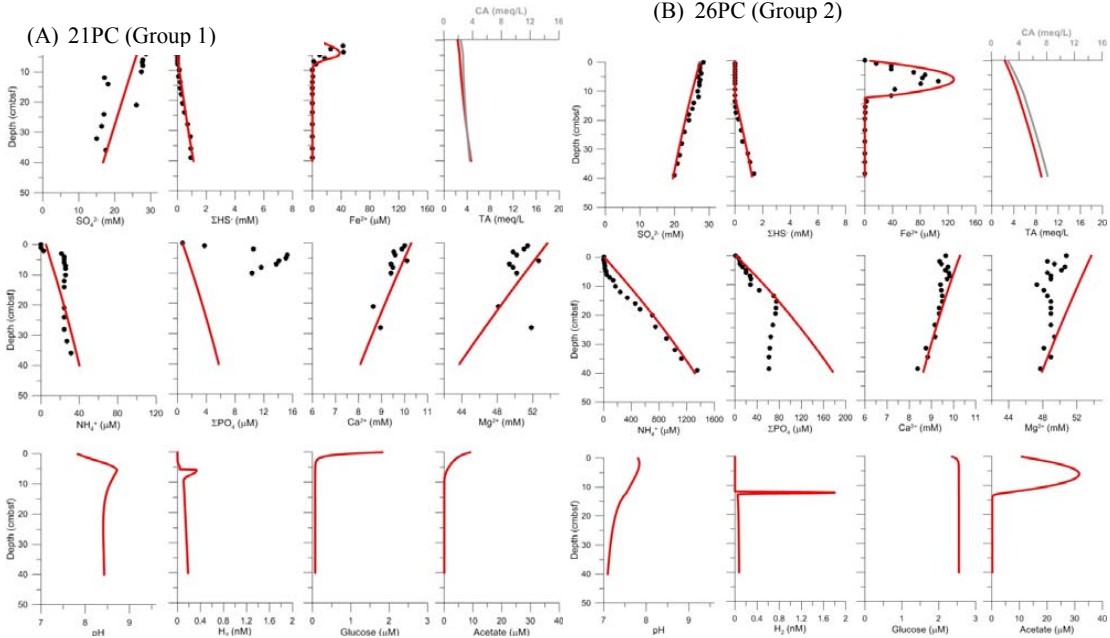

Fig. 5: Model calibration with available porewater data from the four sites with different degrees of methane vs. organic matter influences.



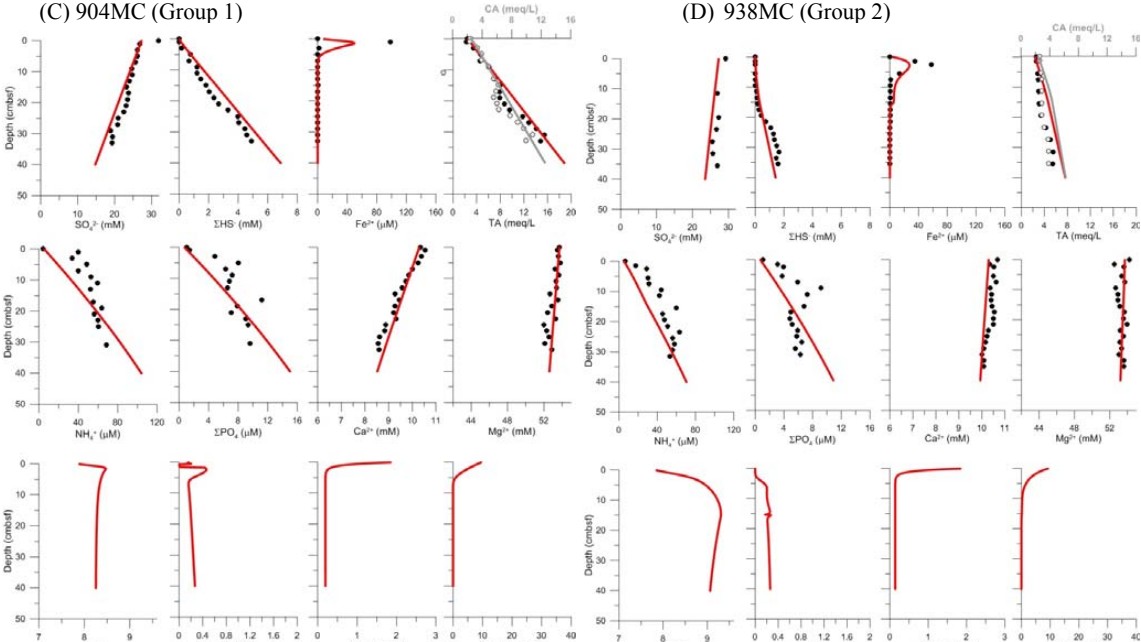





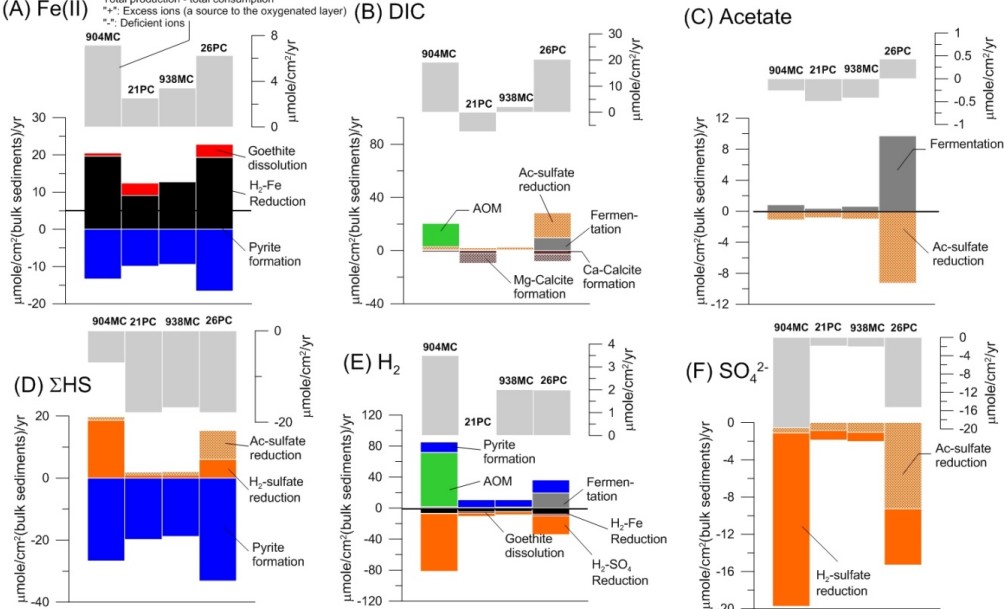

Fig. 6: Model-derived depth-integrated reaction rates for the six chemical species of interest (A)-(F) the first 51cm of modeled sediment column and (G)- (L) the rates in the full sediment column of simulation (151 cm). 904MC and 938MC are from Storfjordrenna GHMs; 21PC is from Hola trough; 26PC is from Ullsfjorden.



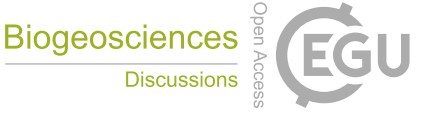

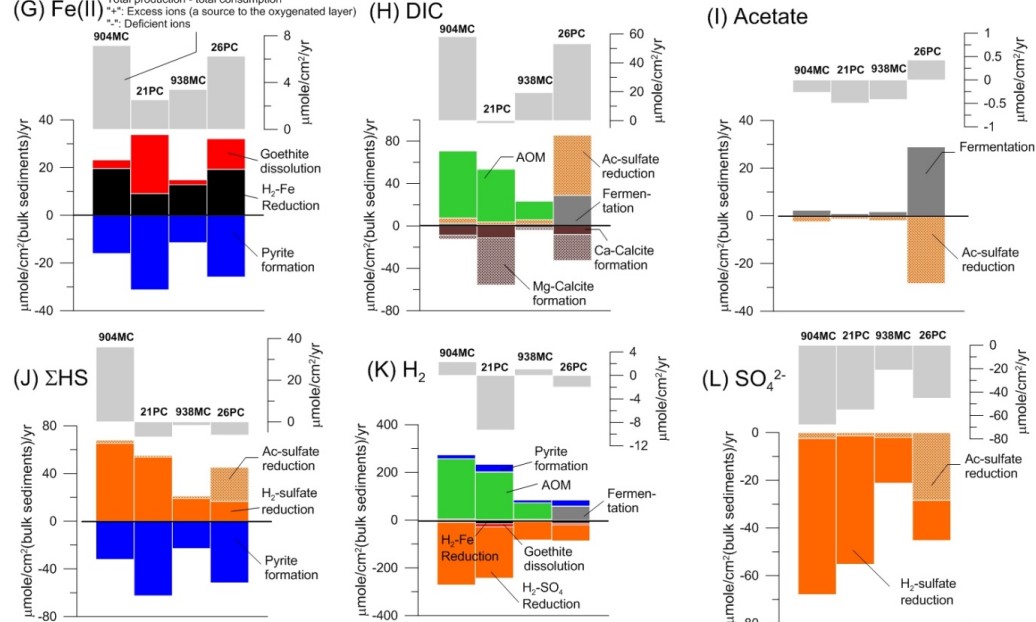

5    By estimating the Fe(II) flux towards the oxic layer in the sediments with Fick's law (Eqs. 3 and 5), we calculated fluxes from 0.03 to 5.06 $\mu mol/cm^2/yr$ (Fig. 7 and column L in Tab. 1). The efflux of Fe(II) into the bottom water varies from 0.01 to 3.73 $\mu mol/cm^2/yr$ (Fig. 7 and column M in Tab. 1). For the sulfate flux, we estimated 1.27 to 175.26 $\mu mol/cm^2/yr$ from the profiles (column K in Tab. 1). In general, high Fe(II) effluxes correspond to high Fe(II) flux towards the oxic layer in the sediments. There is however no obvious correlation observed between sulfate and Fe(II) effluxes.





Fig. 7: Fe(II) fluxes towards (A) oxic sediment layer and (B) bottom water (efflux) estimated from the observed porewater profiles. The percentages of Fe(II) leaving the sediments were shown in (C).



## 5 Discussion

### 5.1 Contrasting drivers of sulfate reduction: AOM vs. organic matter degradation

As sulfate is the most abundant electron acceptor in marine environments, different pathways of sulfate reduction can have a profound impact on the fate of other ions and therefore their distribution in the porewater. It is therefore worth the

effort to have an in-depth investigation into how sulfate is consumed and how the different consumption pathways impact biogeochemistry in shallow sediments. To discuss the interplay among different reactions, we investigated the mass balance of $H_2$ (Figs. 6E&6J and Fig. 8) as it is a crucial substrate involved in the majority of the reactions we discussed. Contrasting sulfate reduction pathways were observed when we compared the depth-integrated rates among the different sites (Fig. 6). Sulfate reduction at most of the sites is driven by HySR, which is fueled by $H_2$ produced from AOM. For example, it is

obvious from Fig. 6K that the rate of $H_2$ production through AOM is very close to the consumption rate of $H_2$ by HySR at 904MC, 938MC, and 21PC, which suggests the tight coupling of AOM and HySR. From 26PC, AOM is insignificant in terms of $H_2$ production. Instead, fermentation (R2 in Fig. 2) plays an important role supplying $H_2$ for HySR:

$$C_6H_{12}O_6 \text{ (aq)} + 4H_2O \rightarrow 2CH_3COO^- + 4H^+ + 4 H_2 + 2HCO_3^-   (8)$$

When we excluded the deepest cell where methane input was assigned and focused only on the shallower sediments (Figs. 6A to 6F), we observed that the overall sulfate reduction rates decreased significantly (Fig. 6F vs. 6L) as most of the reactions are sustained in the deeper sediments through AOM. Increases in HySR rates beneath the iron reduction zone (5-10 cmbsf; Fig. 8) were observed at all four sites with the largest rate observed from 26PC. Such an increase in HySR rate is

likely due to the $H_2$ produced from pyrite formation for 904MC, 21PC, and 938MC (Figs. 8A-8C) and/or from fermentation for 26PC (Fig. 8D). Even though the rate of such HySR is small compared to the rate of sulfate reduction coupled to AOM in deeper sediments (Fig. 6K), this observation suggests a positive feedback in sulfur cycling at the interface where iron reduction and sulfate reduction overlap. A higher flux of porewater sulfide into the base of the iron reduction zone stimulates faster pyrite precipitation which produces more $H_2$ and stimulates faster sulfate turnover. The resulting faster production of

sulfide triggers a positive feedback by accelerating pyrite formation and therefore $H_2$ production. With the presence of labile organic matter, such as the case in 26PC, $H_2$ produced from both pyrite formation and fermentation stimulates a much faster sulfate turnover which also contributes to  the positive feedback.



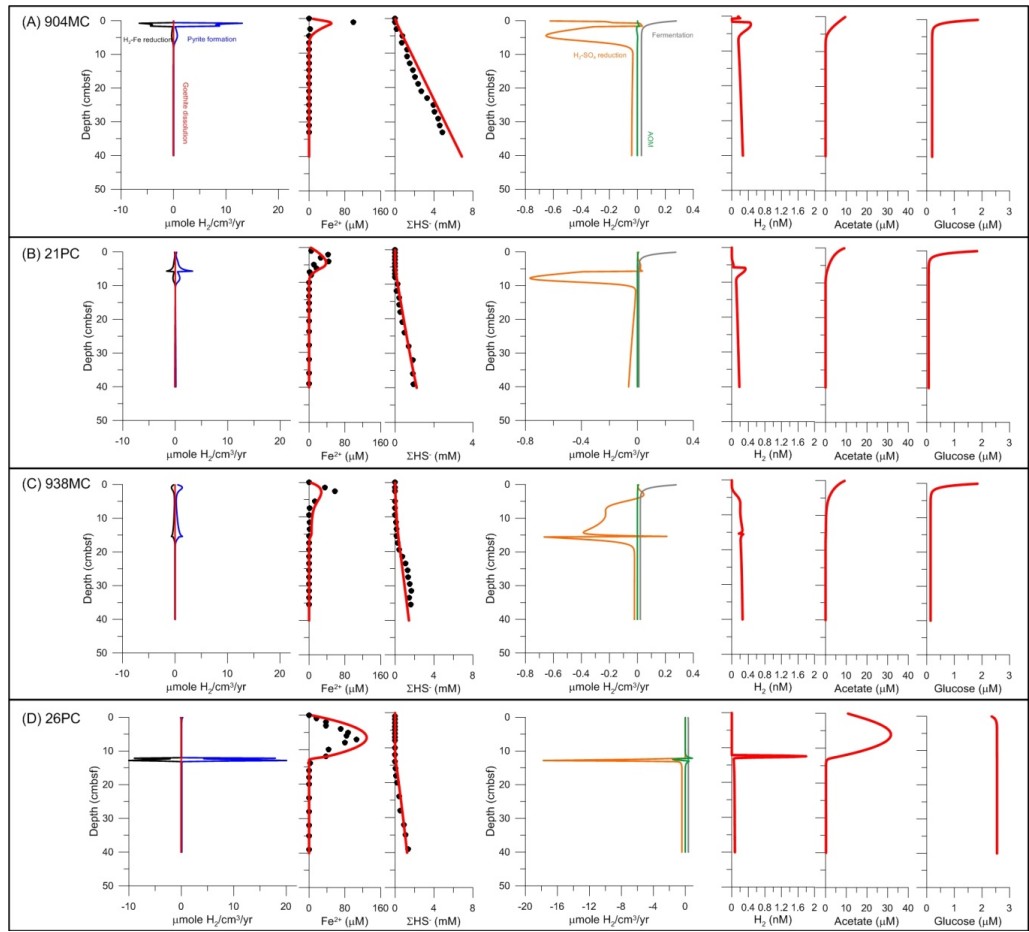

Fig. 8: Model derived rate profiles for $H_2$-associated reactions in the first 51cm of the modeled sediment column.

5      The rapid pyrite formation in  shallow sediments is also confirmed by our solid phase geochemistry data. As shown in
Fig. 4, the contents of CRS increase from ~200 μmole S/g just beneath the sediment-water interface to ten fold at 904MC
and seven fold at 1521GC in just 10 cm. Even though there is no CRS data from 938MC, the increase in the S/Al ratio in the
first 10 cm also suggests rapid formation of iron sulfide minerals. $H_2$ produced through such rapid pyrite formation in the
shallow sediments is more likely to escape the sediment column and diffuse into the bottom water. Indeed, as our $H_2$ mass

10   balance calculation suggests (Fig. 6E), there is a small excess production of $H_2$ (2-4 μmole/cm$^2$/yr) that can potentially
diffuse towards the seafloor, where $H_2$ concentration is low. This is also apparent from the concentration gradient of our





modeled $H_2$ profile shown in Fig. 5. We therefore speculate that the excess $H_2$ produced from pyrite formation and/or fermentation in the shallow sediments can potentially be an important $H_2$ source to sustain the microbial community across the sediment-water interface in methane rich environments.

We also observed different responses of mass balance in DIC when comparing the sites with more AOM-coupled

sulfate reduction (904MC, 938MC, and 21PC) with the site with dominantly organoclastic sulfate reduction (26PC). Stoichiometrically, reduction of one mole of sulfate can produce one mole of bicarbonate ions when coupled to AOM and two moles of bicarbonate ions in the case of coupling with organic matter degradation (R2+R4 vs. R8+R4 from Fig. 2). Such differences explain the similar excess in DIC between 904MC and 26PC (9% higher in 904MC) despite the higher overall sulfate reduction rate in 904MC (20% higher in 904MC) (Fig. 6H). From the sites with AOM-coupled sulfate reduction (e.g.,

904MC), much of the excess DIC can diffuse towards the sediment-water interface and increases the alkalinity or buffer capacity in the bottom water. This is however not the case for sites with prominent organoclastic sulfate reduction. For example, we observed a larger concentration gradient in the carbonate alkalinity (CA) profile from 904MC but a much smaller gradient from 26PC (Fig. 5), despite the similar production rate of excess DIC of the two sites (Fig. 6H). This discrepancy can be explained by the lower porewater pH at 26PC due to fast fermentation. Eq. (8) states that for every mole

of DOC fermented, total alkalinity decreases by two milliequivalents due to higher proton production. We therefore argue that, in contrast to locations with intensive organic matter degradation, cold seep environments are a potentially important source of alkalinity to the overlying bottom ocean water, even though the contribution on a global scale and its impact to ocean chemistry are not yet quantified.

**5.2 Interaction between Fe(II) and ΣHS**

Rapid turnover of sulfate from the four investigated sites produces large amounts of hydrogen sulfide which impacts iron cycling in the surficial sediments. We compared our results with the porewater Fe(II) measurements in Svalbard fjords reported by Wehrmann et al. (2014) (see Fig. 1 for site locations), where much of the sediment is not influenced by methane. Terrestrial surface rocks rich in iron minerals are eroded by glacial activity and transported into the fjords. This additional

oxidized iron source fuels rapid iron reduction and results in high porewater Fe(II) concentration observed in the fjord sediments (Wehrmann et al., 2014). A broad iron reduction zone (> 80 cm) can be observed from the Fe(II) profiles of their sites in the Kongsfjorden and Van Keulenfjorden with very little sulfate reduction as seen from the almost constant downcore sulfate concentration. For the sites in Smeerenburgfjorden, a small but significant amount of sulfate reduction was observed from both the decrease in sulfate concentration paired with increases in sulfide concentration (Wehrmann et al.,

2014). Such sulfate reduction was attributed to organoclastic sulfate reduction by the authors.

Both the fluxes towards the sediment oxic layer and across the sediment-water interface were plotted against water depths at the various locations (Fig. 7). A positive correlation was observed between the water depth and Fe(II) flux towards the oxygenated sediment layer from the dataset in Svalbard fjords (except for two data points; Fig. 7A) which was noted by



Werhmann et al. (2014) and was attributed to the seaward increase of sedimentation rate and input of marine organic matter. The fluxes derived from our study sites, however, depart from this relationship (Fig. 7A), suggesting organic matter input was not the primary control of the iron reduction rate and Fe(II) flux towards the oxic layer at our sites. Both the porewater Fe(II) concentrations are lower (<208μM) and iron reduction zones are thinner (< 10cm except for 53PC; Fig. 3) at our study

sites which suggest the processes involving Fe(II) are more dynamic. The concentrations of Fe(II) and the extents of the iron reduction zone are essentially controlled by both the Fe(II) availability through iron reduction and the consumption of Fe(II) (such as by pyrite formation). In settings with low rates of sulfate reduction and thus production of sulfide (e.g., most of the sites in Wehrmann et al., 2014 ), high porewater Fe(II) concentration and broad iron reduction zones reflect very little Fe consumption; the Fe(II) flux is therefore controlled entirely by the rate of Fe(II) production. On the other hand, when higher

concentrations of sulfide appear in the deeper sediments, due to the intensified sulfate reduction, precipitation of iron sulfide minerals at the bottom of the iron reduction zone will initially decrease the concentration of Fe(II) which will result in a thinner iron reduction zone. In response to such intensified Fe(II) consumption, higher iron reduction is expected as Fe(II) is being removed from porewater. Consequently an increase in Fe(II) concentration within the thin iron reduction zone may lead to a higher Fe(II) flux towards the surface sediments. The interplay between sulfate reduction, iron reduction, and pyrite

formation is further investigated in the next section.

Despite the much lower Fe(II) concentration detected at our sites, the Fe(II) fluxes derived are similar to the estimation from Wehrmann et al. (2014) (Fig. 7A). Such results partially reflect the dynamic interplay between Fe(II) production and consumption as described in the previous paragraph. The Fe(II) flux across the sediment-water interface (efflux) estimated from our sites is also similar or even slightly higher compared to the fluxes from Wehrmann et al. (2014) (Fig. 7B), which is

also reflected by the higher percentages of Fe(II) escaping from the sediment column (Fig. 7C). This is potentially due to the very thin oxic layer above the iron reduction zone as a result of the 'compressed' redox boundary in response to the high sulfide concentration. The acceleration of iron reduction and production of Fe(II) shoals the redox boundary and increases the likelihood of exposing layers with abundant Fe(II) to the bottom water. Once the Fe(II) escapes the sediment column, it is rapidly oxidized and forms nanoparticulate ferrihydrite under oxic bottom water conditions (Raiswell and Anderson 2005)

which will form aggregations or be scavenged by suspended sediments. Such a reoxidation process can happen repeatedly but a significant fraction of either the Fe(II) or freshly formed ferrihydrite can be transported laterally to deeper ocean basins. It has been shown that the reoxidation cycle of reactive iron is the primary driver for the shuttling of reactive iron from the oxic shelf to euxinic basin environments. It was also pointed out that, such an iron shuttle is intensified at locations where the chemocline, the boundary between oceanic anoxic and oxic conditions, intercepts the seafloor (Lyons and Severmann

2006), as Fe(II) can be transported directly without going through the repeated reoxidation. Most of our sites resemble such a condition as inferred from the elevated Fe(II) concentrations in very shallow sediment depth (Fig. 3).

Fe(II) consumption through the precipitation of iron sulfide minerals is also much higher at our sites compared to those from Wehrmann et al. (2014), which is reflected by the high abundance of both CRS and AVS (Fig. 4). At 1521GC, the wt% of AVS-Fe is the highest (0.9 wt% or 154 μmole AVS-S/g sediments) at 3 cmbsf and then gradually decreases to less than



0.3 wt% at 30 cmbsf, which is 3 to 10 times higher than the values from Wehrmann et al. (2014). The highest CRS-Fe abundance from Wehrmann et al. (2014) is no more than 0.25 wt% whereas, from both 904MC and 1521GC, we have measured more than 5 wt% of CRS-Fe. As we noted in the previous section, most of the pyrite formation seems to occur shallow in the sediment (< 10 cmbsf) (Fig. 4), there is however no significant increase of CRS at the depth of the SMTZ (ca.

70 cmbsf) at 1521GC, where rapid formation of pyrite is usually observed (Lim et al., 2011). Such insignificant pyrite formation around the depth of the SMTZ can be explained by the lack of a Fe(II) source deep in the sediments. As our Fe(II) mass balance suggested (Figs. 6A&6G), most of the Fe(II) for pyrite formation comes from reduction of iron (oxyhydr)oxides minerals in the surficial sediments at our sites. Only when all the iron (oxyhydr)oxides minerals deposited on the seafloor are fully consumed by iron reduction, other less reactive iron minerals, such as magnetite, can react to slowly

sustain Fe(II) for pyrite formation deeper in the sediment. Our observation is not unique. For example, it has been observed in Black Sea sediments that, the first peak in CRS content can be as shallow as <50 cmbsf, or 150cm above the SMTZ defined by the porewater profiles (Jorgensen et al., 2004). Such results challenge the applications of inferring the depth of the paleo SMTZ by the distribution of pyrite (van Dongen et al., 2007; Lim et al., 2011). We argue that such an application fails the purpose if most of the pyrite formation occurs close to the seafloor, such as in the case in 904MC and 1521GC.

### 5.3 Fe(II) flux response to fluctuations in methane and seafloor iron supplies- a model perspective

By comparing the porewater profiles and model results from the various sites around Svalbard and northern Norwegian shelf/fjord settings, we have documented the dynamic responses of Fe(II) cycling near the sediment-water interface to the different sulfate reduction pathways. To systematically discuss the variation of Fe(II) flux towards the oxic sediment layer

observed at the various sites (Fig. 7), we conducted model sensitivity tests by 1) assigning different amounts of iron (oxyhydr)oxides on the seafloor for scenarios in which sulfate is predominately consumed by AOM or organic matter and 2) assigning different fluxes of methane in the sediments. We examined the response of the entire reaction network to determine the major factors that influence the Fe(II) fluxes towards the oxic sediment layer from our investigated sites. We used the setup and kinetic constants fitted for 904MC (Figs. 5C and 8A) and 26PC (Figs. 5B and 8D) for the two sensitivity

tests. We focused specifically on the interaction between C, S, and Fe and compared the Fe(II) and sulfate fluxes derived from our porewater profiles (Tab. 1) with the rates derived from the modeling.

In the first sensitivity test, we applied the model setups from 904MC and 26PC, where methane and organic matter are the dominant drivers of the reaction network (Fig. 9). We observed two major impacts on the cycling of iron and sulfur due to such disturbance. First, pyrite forms faster initially due to the higher HyFeR rate and greater availability of Fe(II)

(between 2.3 to 2.4% of iron (oxyhydr)oxides in Fig. 9A and 1.35 to 1.45% of iron (oxyhydr)oxides in Fig. 9C). Even though sulfate reduction is expected to accelerate due to the removal of sulfide through pyrite formation, the total sulfate reduction rate does not increase as fast as HyFeR due to the inhibition from higher porewater Fe(II) concentrations (grey lines in Figs. 9B&9D). This is especially obvious in the case when organic matter is the primary driver for sulfate reduction (e.g., 26PC in Fig. 9D). The other results from our sensitivity test is, even though HyFeR is accelerated due to the increasing



iron (oxyhydr)oxide deposition, dissolution of $Fe_{LR}$ is suppressed as iron (oxyhydr)oxide is more thermodynamically favorable between the two mineral phases. This effect will have great impact if $Fe_{LR}$ contributes significantly to the formation of pyrite and therefore can counter-balance the increasing Fe(II) production from HyFeR (e.g., 26PC in Fig. 9C).

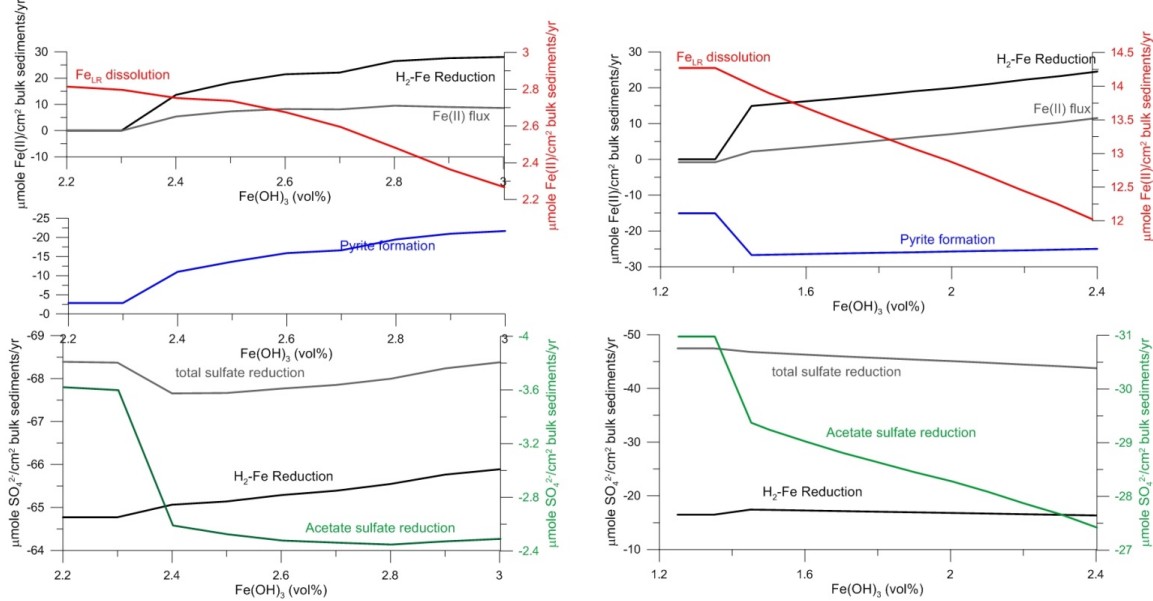

Fig. 9: Model sensitivity tests with variable amount of iron (oxyhydr)oxides on the seafloor. We tested two scenarios: (A&B) sulfate reduction is coupled with AOM and (C&D) sulfate reduction is coupled with organic matter degradation. As expected, more iron on the seafloor results in higher iron reduction and pyrite formation rates. However, higher concentration of Fe(II) also inhibits sulfate reduction which diminish sulfide production and poses a negative feedback to the pyrite formation.

In the second sensitivity test, in which we varied the supply of methane in the sediments, we observed a complicated relationship between Fe(II) flux and total sulfate reduction rate (i.e., sum of AcSR and HySR rates) (Fig. 10A). The reaction network undergoes three different stages as methane flux increases. In the first stage (total SR rate <40 μmol/cm²/yr), Fe(II) flux increases steadily and monotonically with increaing SR rate. In this stage, sulfide flux is in general very low as high

10 production rates of Fe(II) through both HyFeR and $Fe_{LR}$ dissolution limit the concentration of sulfide through pyrite formation (Fig. 10B). This imples that, the rate of pyrite formation during this stage is controlled by the availability of sulfide. The fluxes derived from most of our sediment cores (10 out of 12) fall in this range. We observed a first-order agreement between the porewater profile-derived fluxes and our second model sensitiviy test as shown in Fig. 10A. The scatterness suggests that  seafloor iron (oxyhydr)oxide deposition is still an important factor in determining the flux of Fe(II)




as we demonstrated in the first sensitivity test (Fig. 9). However, an intensifying methane flux can modulate the Fe(II) flux towards the oxic sediment layer at a specific site where seafloor iron (oxyhydr)oxide input is not a varible.

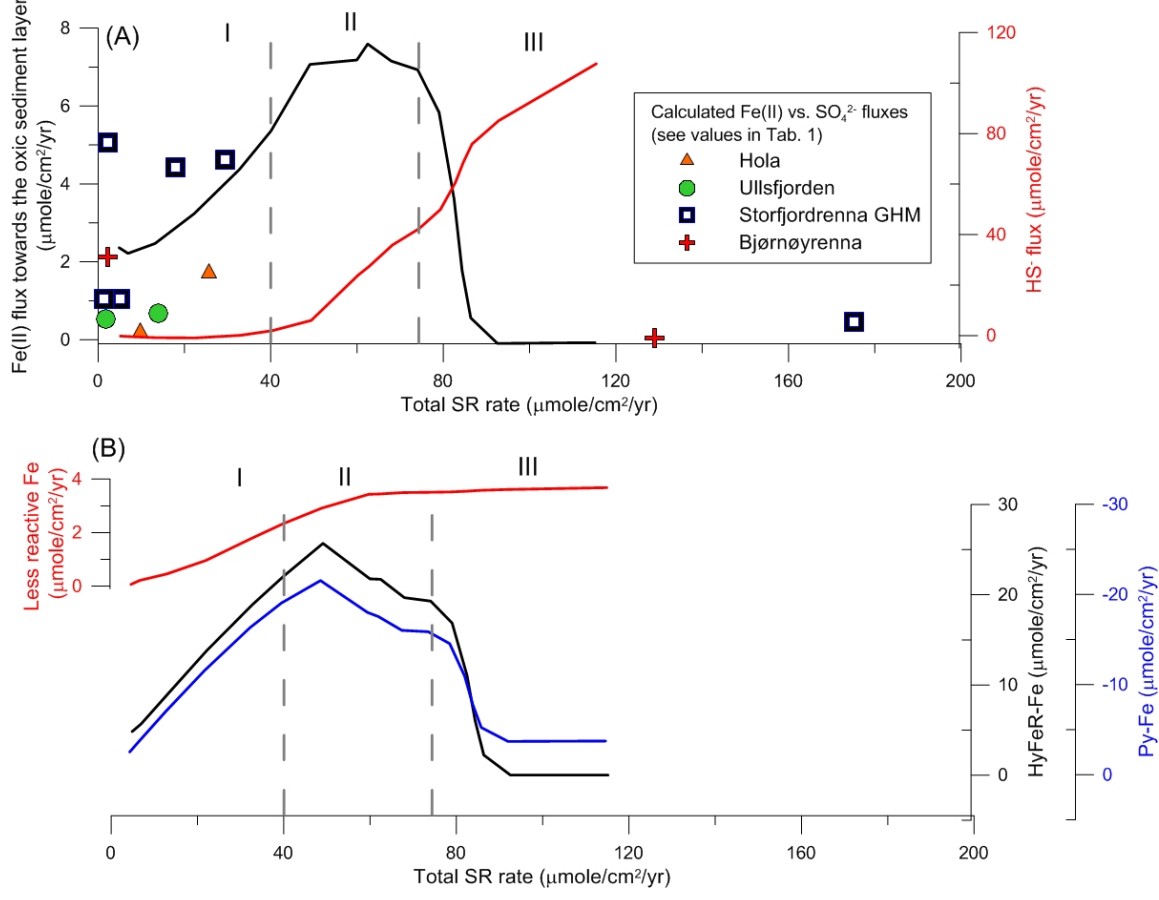

Fig. 10: A model sensitivity test with the variable methane supply (model setup from 904MC). (A) Comparison of fluxes of Fe(II) and $\Sigma$HS in the sediments derived from the modeling (lines) and porewater profiles (dots) with respect to total sulfate reduction (SR) rate, which is the sum of HySR and AcSR rates. (B) Variation in rates for pyrite formation, dissolution of $Fe_{LR}$, and iron (oxyhydr)oxide dissolution with different total sulfate reduction rates derived from the model.

In the second stage (total SR rate between ca. 40 to 75 $\mu$mol/cm$^2$/yr), the Fe(II) flux remains high (ca. 7 $\mu$mol/cm$^2$/yr) while the sulfide flux increases at a moderate slope (Fig. 10A). During this stage, both the rates of HyFeR and pyrite



formation decrease with increasing total SR rate (Fig. 10B). Pyrite formation rate is proportional to but lower than HyFeR rate, as in the first stage, indicating the coupling of these two reactions. The increase in $Fe_{LR}$ dissolution rate continues but turnes to a milder increase when the Fe(II) flux reaches its maximum value (total SR rate at ca. 60 μmol/cm$^2$/yr). Fe(II) production is obviuosly not able to cope with the much faster sulfide production in this stage and results in a rapid increase

in sulfide flux towards the seafloor. However, as there is still Fe(II) in the porewater, very little of the sulfide reaches seafloor. Even though no flux derived from our porewater profiles falls in this range, the condition from 904MC (Figs. 3A and 5C) may, to the first degree, resemble the situation in this stage. ΣHS concentration is detectable already at 3 cmbsf with a very rapid increase to 4.9 mM at 33 cmbsf. Even though the sulfide efflux is still very low as we were not able to detect any sulfide above 3 cmbsf, one would expect a sharp increase in sulfide efflux if sulfate reduction is slighlty elevated.

In the last stage (total SR rate is higher than 75 μmol/cm$^2$/yr), a rapid decline in Fe(II) flux is accompanied by a rapid increase in sulfide flux towards the seafloor. The rate of HyFeR essentially drops to zero as the rapid turnover soon exhausts all iron (oxyhydr)oxide minerals by the end of our simulation (1500 years). As HyFeR is the most important Fe(II) source for pyrite formation, the drop in pyrite formation rate is expected. During this stage, pyrite formation is essentially limited by the availability of Fe(II), which is entirely from $Fe_{LR}$ dissolution (Fig. 10B). Two sediment cores recovered from bacterial

mats by ROV-operated push corer and blade corer (1029PC and 1123BLC from Fig. 3C) nicely demonstrate the case during this stage. We were able to detect ΣHS (ca. 20 μM) from the samples collected in coretop seawater from 1123BLC. Rapid increases to 4.97 mM at 13cmbsf and over 6 mM only 1 cmbsf from 1029PC and 1123BLC, respectively, reveal a very high flux of sulfide towards the seafloor or even to the bottom water. The higheset Fe(II) concentrations are only 10 and 1.7 μM from the two cores (Fig. 3D) which results in a very low calculated Fe(II) flux towards both the oxic sediment layer (Fig.

10A) and bottom seawater (Fig. 7C).

### 5.4 Correlating the distribution of chemosynthesis based animals with fluxes of Fe(II) and hydrogen sulfide

As the abundance of Fe(II) in the shallow sediments determines how much hydrogen sulfide can reach the seafloor (Fig. 10A), this flux therefore influences the porewater Fe(II) chemistry and the distribution of chemosynthesis based bio-

communities on the seafloor. In order to investigate this correlation, we compared seafloor observations with the representative downcore Fe(II) and ΣHS profiles from the three stages in our second sensitivity test as shown in Fig. 11. The first stage of medium Fe(II) and low ΣHS fluxes (Fig. 11D), corresponds to the area from where core 938MC was recovered that is devoid of any chemosynthesisbased animals but colonized by non-seep specific fauna such as *Thenea* sponges and anemones (Fig 11A). These animals lack the adaptations that seep specific fauna have evolved in order to combat the toxic

effects of their sulfidic habitats, and their presence on the seafloor demonstrates the absence of sulfide leaking up to the bottom water.

In the second stage at which both Fe(II) and ΣHS fluxes are higher than the first stage (Fig. 11E), a dense distribution of *Oligobrachia* worms can be observed close to the location where 904MC was recovered (Fig. 11B). *Oligobrachia* worms are



frenulate sibognlid polychaetes and have been reported  from both Storfjordrenna GHMs and Bjørnøyrenna (Sen et al., in prep). They are thin (500 µm diamater) and long (50-60 cm) worms that contain sulfur oxidizing bacterial symbionts in their trunks (Sen et al., 2018). The majority of the worm lies within the sediment and they transfer dissolved reduced sulfur compunds, such as hydrogen sulfide and thiosulfate, across their thin tubes and epidermis. Based on their distribution related

to different gas seepage regimes, Sen et al. (2018) hypothesized that, besides adequate concentration of sulfide, high supply or flux of sulfide is also important  Frenulates sibogliind worms were also hypothesized to be capable of utilizing iron sulfide minerals (Dando et al., 2008). However, the particular species present at the Storfjordrenna GHMs and Bjørnøyrenna is considered to differ from other frenulates, in that it relies almost exclusively on dissolved sulfide (Sen et al., in prep).

In the third stage, at which iron (oxyhydr)odxide dissolves immediately upon deposition due to  high sulfide flux (Fig.

11C), *Olgiobrachia* worms are also abundant which corresponds to the location where 1123BLC was recovered (Fig. 11C). During this stage, sulfide is able to reach the bottom water as shown by our measurements (Fig. 3C). Consequently, filamentous sulfur oxidizing bacteria were seen to densely colonize the anterior ends of *Oliogbrachia* tubes that extend a few centimeters into the water column, resulting in a fuzzy appearance as seen from Fig. 11C (or from the image of better resolution in the appendix). It is also noteworthy that, immediately adjacent to the bacteria covered worms, microbial mats

were present but not the *Olgiobrachia* worms (the area with marks of blade corer in Fig. 11C). We speculate that the sulfide flux in this field of bacterial mats is even higher and results in a localized sulfidic bottom water condition. As *Oliogbrachia* worms and their symbionts still require oxygen for survival, which is why their anterior ends are extended to the water a few centimeters above the anoxic sediments, the localized sulfidic bottom water may interfere with the oxygen uptake of the worms and explain their absence. Such an explanation is however complcated by the fact that the bacteria constitute the mats

are also aerobic and require access to oxygen. Future studies target the two habitats are required to test our speculation.



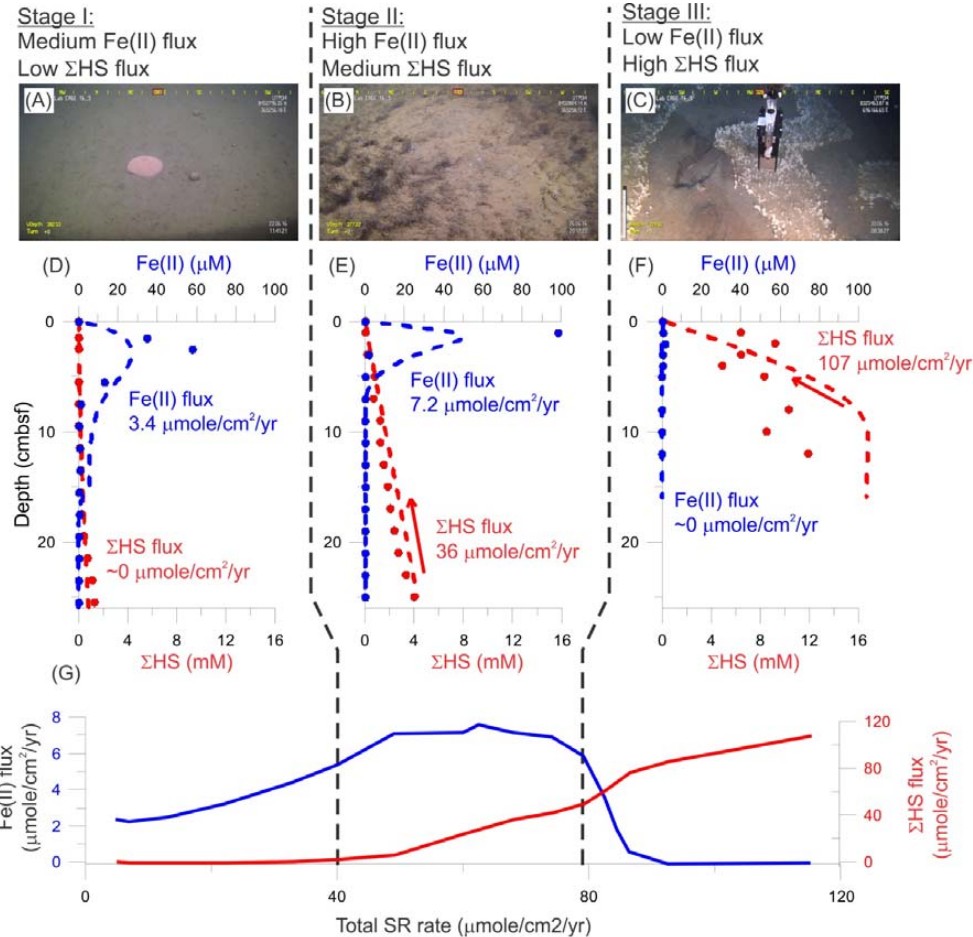

Fig. 11: Relationships between seafloor characteristics (A-C) and porewater profiles (D-F) for the three stages in our second model sensitivity test. For scales in each panel, the anemone in (A) is ca. 10 cm in diameter, the worm is 3 cm long, and the blade core in 20 cm by 10 cm. All fluxes in the figure refer to the flux towards the oxic sediment layer. In the first stage at which the Fe(II) flux is high and sulfide flux is very low, seafloor is devoid of any chemosynthetic-based animals. In the second stage, due to the thinner iron reduction zone, the *Oliogbrachia* worms are able to harvest the hydrogen sulfide and sustain the bacteria inside the worms. In the third stage, the high flux of hydrogen sulfide reaching the bottom water allows the sulfur oxidizing bacteria to grow on the anterior ends of the tubes. Enlarged version of (A) to (C) is included as supplementary material.





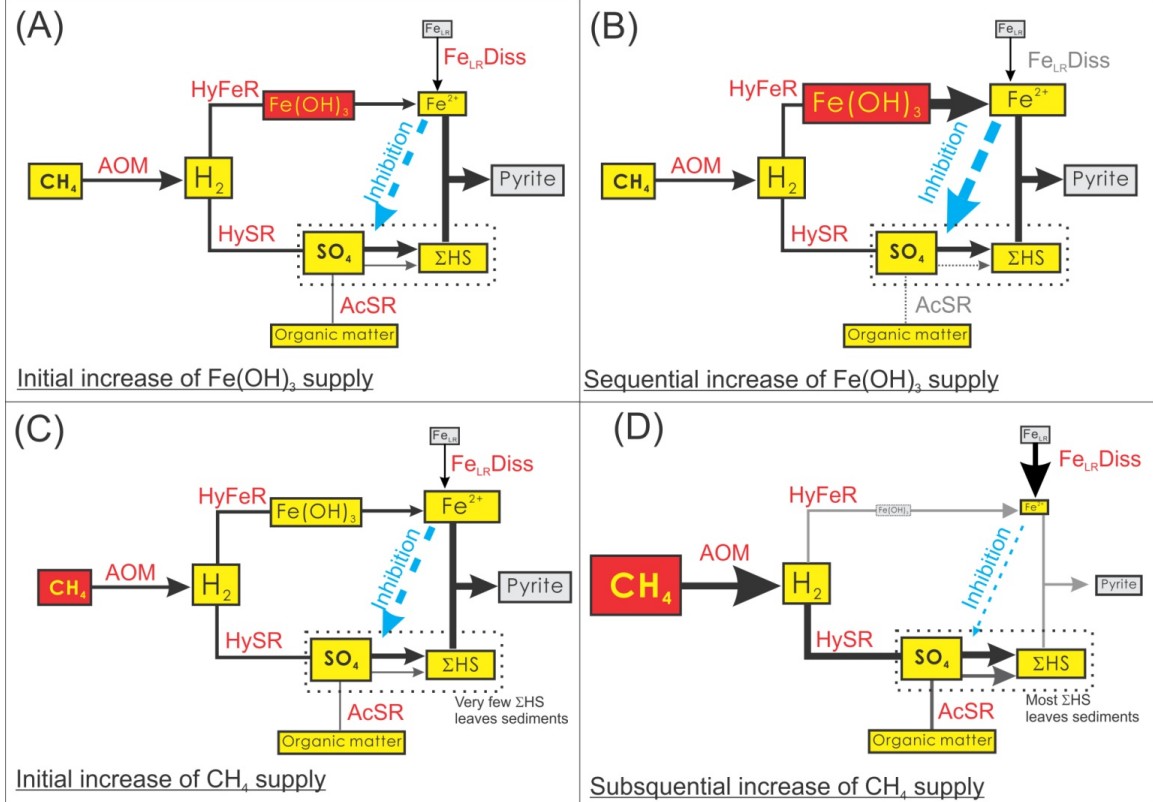

Fig. 12: A summary of expected responses from the reaction network with different disturbances. (A) With higher seafloor iron (oxyhydr)oxide deposition, pyrite formation is initially stimulated. Higher Fe(II) concentration in the porewater poses a negative feedback to sulfate reduction and therefore the production of hydrogen sulfide. (B) With further increase in seafloor iron (oxyhydr)oxide deposition, pyrite formation is less encouraged due to the more intense inhibition of sulafte reduction from Fe(II). Such inhibition is especially obvious when organic matter is the primary driver of sulfate reduction. (C) With an intial increase of methane supply, both sulfate reduction and iron reduction are stimulated due to the higher rate of pyrite formation. (D) With an even larger supply of methane, extrememly fast sulfide production can exhaust all iron (oxyhydr)oxide and results in a termination of pyrite formation.





## 6 Conclusions

We present porewater data from four recently investigated cold seeps in the Arctic Ocean and quantitatively investigated the porewater data by calculating fluxes and performing numerical modeling. Our results reveal complicated feedbacks among sulfate reduction, pyrite formation, and iron reduction. Even though the Fe(II) fluxes towards the oxic

surface sediments from our methane-influenced sites are slightly lower than those at background sites reported in Wehrmann et al. (2014), proportionally more Fe(II) is able to reach the bottom ocean water from our sites due to the more confined and shallower iron reduction zones. We also showed that cold seeps could be potentially important sources of alkalinity to the ocean. Besides, the hydrogen gas produced during pyrite formation immediately beneath the thin iron reduction zone from the sulfide-rich sediments could have a significant impact on the biogeochemistry of shallow sediments. The high pyrite

abundance right beneath the iron reduction zone and the absence of a CRS peak at the depth of the SMTZ argue against the use of pyrite abundance as an indication of the SMTZ in the past.

With our data-constrained transport-reaction model, we investigated the interplay between different reactions by assigning more seafloor iron (oxyhydr)oxide or higher methane supply in the sediment column. When there is higher seafloor deposition of iron (oxyhydr)oxide, HyFeR is immediately accelerated but the decompositon of other less reactive

iron minerals is now suppressed (Fig. 12A). Pyrite formation is strongly favored during the initial increase of seafloor iron (oxyhydr)oxide but is less enhanced hen even higher amounts of iron (oxyhydr)oxide are deposited on the seafloor, which is due to the inhibition of sulfate reduction by higher concentrations of porewater Fe(II) (Fig. 12B). This is especially obvious when organic matter is the primary driver of sulfate reduction (i.e., AcSR).

When the system is disturbed by a higher supply of methane in the sediment column, faster sulfate reduction and

therefore sulfide production are the immediate consequences. More Fe(II) is removed from the porewater through pyrite formation which initially accelerates HyFeR by removing the product of the reaction (i.e., Fe(II)) (Fig. 12C). At a certain sulfate reduction rate, the reduction of iron (oxyhydr)oxide reaches its highest rate and can no longer stop hydrogen sulfide from reaching the seafloor. With a continuously increasing methane flux (and therefore sulfate reduction rate), all seafloor iron (oxyhydr)oxide immediately dissolves upon deposition. The rate of pyrite formation is greatly reduced due to the

overwhelming supply of sulfide and inproportionally low Fe(II) supply. During this stage, hydrogen sufide can freely escape to the seafloor without being sequestrated by pyrite formation (Fig. 12D).

Such complicated feedbacks among the different biogeochemical reactions have a profound impact on the seafloor animals that rely on hydrogen sulfide as their energy source. This was clear when we correlated our model results and porewater profiles with seafloor observations. When sulfide production is very low and the iron reduction zone is thick

(>50cm), sulfide at the seafloor becomes limiting and there are no bacterial mats and chemosynthesis based animals established on the seafloor. With  faster sulfate reduction and a much narrower iron reduction zone (e.g., 10-15cm thick), chemosynthesis based animals such as *Oligobrachia* worms are able to reach beneath the relatively thin iron reduction zone and harvest hydrogen sulfide. With an even faster sulfate reduction, in which case there is essentially no iron reduction due




to the extremely fast sulfide production, the high concentration of sulfide reaching seafloor not only promotes the colonization of the *Oligobrachia* worms but also encourages the growth of white sulfur-reducing bacteria on the parts of these worms that extend a few centimeters above the sediment surface.

## 7 Figure captions

Fig.1: Map showing the four Arctic cold seeps discussed in this paper. Sites with available data for sedimentary sulfur extraction and magnetic susceptibility were labeled in yellow.

Fig. 2: The reaction network assigned in our transport-reaction model. See text and Tab. A1 in the Supplementary material for details.

Fig. 3: Porewater profiles of the 12 investigated sediment cores (excluding 1521GC). We categorized them to: (A) sites with
significant methane influence; (B) sites with little methane influence and higher contribution from organic matter degradation at some sites; (C) sites with the highest methane influence but also under a non-steady-state condition. See text for the criteria of our categorization. CA: carbonate alkalinity; TA: total alkalinity.

Fig. 4: The downcore variations in acid-volatile sulfur (AVS) and chromium reducible sulfur (CRS) abundance as well as the magnetic susceptibility (MS) and XRF-derived S/Al ratios at the three investigate sites from Storfjordrenna GHMs.

Fig. 5: Model calibration with available porewater data from the four sites with different degrees of methane vs. organic matter influences.

Fig. 6: Model-derived depth-integrated reaction rates for the six chemical species of interest (A)-(F) the first 51cm of modeled sediment column and (G)- (L) the rates in the full sediment column of simulation (151 cm). 904MC and 938MC are from Storfjordrenna GHMs; 21PC is from Hola trough; 26PC is from Ullsfjorden.

Fig. 7: Fe(II) fluxes towards (A) oxic sediment layer and (B) bottom water (efflux) estimated from the observed porewater profiles. The percentages of Fe(II) leaving the sediments were shown in (C).

Fig. 8: Model derived rate profiles for $H_2$-associated reactions in the first 51cm of the modeled sediment column.

Fig. 9: Model sensitivity tests with variable amount of iron (oxyhydr)oxides on the seafloor. We tested two scenarios: (A&B) sulfate reduction is coupled with AOM and (C&D) sulfate reduction is coupled with organic matter degradation. As
expected, more iron on the seafloor results in higher iron reduction and pyrite formation rates. However, higher concentration of Fe(II) also inhibits sulfate reduction which diminish sulfide production and poses a negative feedback to the pyrite formation.

Fig. 10: A model sensitivity test with the variable methane supply (model setup from 904MC). (A) Comparison of fluxes of Fe(II) and $\Sigma HS$ in the sediments derived from the modeling (lines) and porewater profiles (dots) with respect to total
sulfate reduction (SR) rate, which is the sum of HySR and AcSR rates. (B) Variation in rates for pyrite formation, dissolution of $Fe_{LR}$, and iron (oxyhydr)oxide dissolution with different total sulfate reduction rates derived from the model.





Fig. 11: Relationships between seafloor characteristics (A-C) and porewater profiles (D-F) for the three stages in our second model sensitivity test. For scales in each panel, the anemone in (A) is ca. 10 cm in diameter, the worm is 3 cm long, and the blade core in 20 cm by 10 cm. All fluxes in the figure refer to the flux towards the oxic sediment layer. In the first stage at which the Fe(II) flux is high and sulfide flux is very low, seafloor is devoid of any chemosynthetic-based animals. In the second stage, due to the thinner iron reduction zone, the *Oliogbrachia* worms are able to harvest the hydrogen sulfide and sustain the bacteria inside the worms. In the third stage, the high flux of hydrogen sulfide reaching the bottom water allows the sulfur oxidizing bacteria to grow on the anterior ends of the tubes. Enlarged version of (A) to (C) is included as supplementary material.

Fig. 12: A summary of expected responses from the reaction network with different disturbances. (A) With higher seafloor iron (oxyhydr)oxide deposition, pyrite formation is initially stimulated. Higher Fe(II) concentration in the porewater poses a negative feedback to sulfate reduction and therefore the production of hydrogen sulfide. (B) With further increase in seafloor iron (oxyhydr)oxide deposition, pyrite formation is less encouraged due to the more intense inhibition of sulafte reduction from Fe(II). Such inhibition is especially obvious when organic matter is the primary driver of sulfate reduction. (C) With an intial increase of methane supply, both sulfate reduction and iron reduction are stimulated due to the higher rate of pyrite formation. (D) With an even larger supply of methane, extrememly fast sulfide production can exhaust all iron (oxyhydr)oxide and results in a termination of pyrite formation.

## 8 Code availability

All software routine can be made available by contacting W.-L. H.

## 9 Data availability

All porewater and solid phase data can be found in the Appendices

## 10 Appendices

1) Data table with all available porewater and solid phase data
2) High resolution photos in Fig. 11(A) to (C)

## 11 Author contributions

WH collected and analyzed the samples from Storfjordrenna and Bjørnøyrenna. WH and PL designed the study, ran the modeling, and wrote the majority of the manuscript. PL also performed the analyses of sulfide concentration for some of the



study sites. SS contributed the data from Hola trough and Ullsfjorden. AS contributed to the discussion of geochemistry and biology correlation. WH, WPG, and F.F. performed the sulfur extraction from the sediment samples. AL contributed to the analyses of nutrient concentration.

**12 Acknowledgements**

5  We would like to acknowledge the captains and crews onboard R/V Helmer Hanssen for cruises in April 2013, CAGE15-2, CAGE15-6, and CAGE16-5. We are grateful for the cruise organization by the chief scientists, Dr. Giuliana Panieri, Prof. Dr. Jurgen Mienert, and Dr. Michael Carroll, for the three CAGE cruises. We greatly appreciate the team from Woods Hole Oceanographic Institution (WHOI) MISO (Multidisciplinary Instrumentation in Support of Oceanography) (Dr. Daniel Fornari and Gregory Kurras) operating the TowCam deep-sea imaging system and the ROV team (Prof. Dr. Martin

10  Ludvigsen, Frode Volden, Stein Nornes and Pedro de la Torre) from the Norwegian Centre for Autonomous Marine Operations and Systems (AMOS) for their assistance in sampling collecting. We would also like to thank Mrs. Haoyi Yao for the assistance in porewater sampling and analyses. This work was supported by the Research Council of Norway through its Centres of Excellence funding scheme (project number 223259) and NORCRUST (project number 255150).





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
