# Peer review of "Dynamic interactions between iron and sulfur cycles from Arctic methane seeps"

_Biogeosciences, 2018_

## Referee Comment (RC1) · Anonymous Referee #1 · 22 Jun 2018

The cycling of iron and sulfur in methane-rich sediments at various seep locations near Svalbard is investigated. Pore-water profiles are used to calibrate a reaction-transport model that takes into account intermediate organic matter breakdown species, such as acetate and molecular hydrogen ($H_2$). This approach provides potentially interesting insights into the coupling of various reaction pathways. For instance, they argue that pyrite formation may produce $H_2$ and stimulate sulfate reduction, and that the accumulation of dissolved iron inhibits sulfate reduction. The calibrated model is then used to explore the effects of increasing iron oxide loading and methane influx on the turnover of chemical species and the partitioning between various reaction pathways. The data and model are also used to estimate dissolved iron fluxes to the overlying water.

Although I find the modeling approach very interesting, a number of important reactions are missing. First of all, sulfidic reduction of iron oxides is not included, which is, as the authors acknowledge, important (page 23, line 14-15). For pyrite formation they use a reaction pathway in which H2 is produced. It may be more likely that most pyrite is formed through the FeS + S0 → FeS2 reaction, where FeS can be formed by sulfidic reduction of iron oxides and the Fe(II) + H2S reaction (see the specific comments). From the low bottom-water temperature and the abundance of polychaetes (see paragraph 5.4) one may expect a high influx of oxygen into the sediment, but the model does not include oxygen and re-oxidation reactions. This is likely to have a large impact on the estimated Fe(II) and H2 fluxes to the overlying water. In Fig. 5 there is a rapid decrease of ammonium in the upper 5 cm, which may be caused by nitrification or rapid transport/mixing.

The data does not provide direct constraints on the transport in the model. The authors briefly mention methane seepage, gas bubble irrigation, and bioturbation as potential drivers of transport for the cores in Fig. 3C. In a cold seep one may expect up- or downward water flow (Tryon and Brown, 2004 and references therein), and the high number of large polychaetes makes it likely that also bioirrigation is important. All these flow processes could affect the profiles in Fig. 3A,B as well. The transport may have a large impact on the interpretation of pore profiles, so the authors may want to discuss these processes in more detail.

The results and discussion in the paper could also be better embedded within the context of other studies. Here are some examples that the authors may find interesting to consider: there are papers about reaction-transport models in cold seeps (e.g. Luff and Wallmann, 2003; Luff et al., 2004). There is a paper on carbon, iron, sulfide cycling in methane-rich sediments (Rooze et al., 2016), which may provide information about the importance of re-oxidation reactions and iron solids that are not included in the model. Rather critical I think are papers that discuss dissolved Fe fluxes to the overlying water (see paragraph 4.5 in Raiswell and Canfield, 2012; Dale et al. 2015). These papers stress the importance of aerobic iron re-oxidation and bioirrigation - factors that

are ignored in the model.

In summary my main concerns are that a few important reactions are not included in the model and that the transport is not well constrained. Both factors could have a major influence on estimates of the dissolved iron fluxes. The reactions could be added to the model, and the authors may want to explore and describe the sensitivity of the model towards different transport regimes.

Specific comments:

Page 2:

Line 8: "Redox reactions... carbon oxidized." To which 'order' does this sentence refer? If they refer to the stratification it is important to note that it is not always the case; for instance, bioirrigation, bubble irrigation, and transient diagenesis can mix the sequence up.

Line 18: "a quantitative... still lacking." This does not do justice to the body of literature that already exists and deals with cold seep biogeochemistry.

Line 20: "Previous studies..." Which studies exactly? The reduction of iron oxides also depends on other factors, such as the crystal structure and pH.

Line 24: Pyrite formation The authors focus on one mechanism of pyrite formation, which involves the production of H2. They do not include the $FeS + S0 \rightarrow FeS2$ reaction, where the FeS can be formed by $Fe + H2S + 2 HCO3 \rightarrow FeS + 2 CO2 + 2 H2O$ and $2 FeOOH + 3 H2S \rightarrow 2 FeS + S0 + 4 H2O$. How likely is it that the reaction pathway in which H2 is produced occurs in Arctic sediments? Rickard and Luther (2007) briefly discuss this reaction (see reaction 29 in the article) and indicate that this reaction may only occur at higher temperatures. Can the authors make a case for the use of this particular reaction pathway and leaving out other pathways with FeS as an intermediate? Rickard (2012) provides a good overview of the various mechanisms that can produce pyrite in sediments.

Line 30-32: "It is therefore expected... oxic layer in the sediment." Is this based on literature? Since dissolved Fe and sulfide react rapidly, would it not be more common that either all sulfide or all dissolved Fe is titrated out of the pore water?

Page 3:

Line 7: "bioavailable" The authors make somewhat broad statements that should be backed up with references. Since Fe(III) can actually be taken up by microbes it is also bioavailable (Raiswell and Canfield, 2012).

Line 8: "Soluble Fe(II) ... can only be produced from anoxic environments" This is imprecise. Fe(II) can be produced in the water column, but most of it will be rapidly reoxidized.

Page 5:

Fig 1: The font size is too small.

Page 10:

Line 16: Perhaps it is better nomenclature to call these processes hydrogenotrophic and acetoclastic sulfate reduction.

Line 30-31: "it is mathematically best to assume only the overall reaction" It is not clear to me how this can be proven mathematically.

The explanation of the model is very terse. It is rather difficult to find all the information since the governing equations are not provided, there are no tables that sum up all the state variables, reaction stochiometries, and rate constants, and the information can only be found in other articles. The authors mention Hong et al. (2016) for which there is no entry in the bibliography. The other reference is to a supplementary file from a Nature Communications paper, but it seems only to give information about the reactions, not the transport.

Page 11:

Fig 2: In the figure Co3 should be CO3. Sulfate is negatively charged. It takes effort to find all the labels in the figure, especially since so many different colors are used. For people with poor sight or bad printers it would be nice to have an overview of all the reactions in a table. It is not clear whether the reactions are reversible.

Line 5-10: "For the reactions involving minerals (R1, R3, R9. . .." The paragraph starting here is a bit terse. Why did the authors choose this approach? The way this rate law is written implies that it is reversible, since $(1 - Q/K)$ can be positive and negative. To describe POC hydrolysis as a reversible equilibrium reaction appears troublesome to me.

Page 12:

Line 4-5: "the equilibrium constants for other reactions were calculated. . . assuming 25 C" The actual temperature is ∼1 degree Celsius (table 1) and thus significantly lower than the 25 degree Celsius the authors use. It is a large difference. Why can the equilibrium constants not be corrected for the in-situ temperature?

Line 9-10: "For reactions . . . Monod-type reaction with the basic form of" Why do the authors choose to use this rate expression? It's not clear whether the term (1-Q/Keq) act as an on/off switch or that the reaction is reversible.

Line 19: "we set an imaginary mineral to produce methane. . ." Why do the authors choose this approach?

Line 23: "We used. . . as the initial condition" What are the initial conditions of the solids?

Line 27: "We then assigned the amount of iron hydroxide in the surface sediments (i.e. the upper boundary condition)"

I guess that 'iron hydroxide' should be 'iron oxide'. It is not clear to me what is used as boundary condition for the solids. A table with all boundary conditions would be helpful.

Page 13:

Lines 14-18: "For a… balance requirement" I am wondering how sensitive these fluxes are to computational errors. Would it not be better to calculate the fluxes across the top boundary directly?

Line 26: There should be a minus sign in eq. 3.

Line 29-line 8(page 14): How did the authors calculate the specific values for the diffusion coefficients? Line 29 suggests they were somehow corrected for different temperatures, but in line 32 it is stated that only one value was used. Why is the effect of bioturbation included for dissolved iron but not for sulfate? Middelburg's relationship for $D_b$ is a function of water depth, and is an average from different sites. However, given the rich seep biota it is likely that seeps form outliers with much higher $D_b$ values.

The organization could perhaps be improved by making separate subsections about the reaction network, boundary conditions, and initial conditions. Information on the input of methane and other boundary conditions seems to be missing, and I also cannot find the parameterizations for the different fits. I don't recall reading anything about the advection of dissolved species.

Page 14:

Equations 5 and 6: It is somewhat inconsistent that here the authors account for the re-oxidation of reduced iron, but not in the reaction-transport model.

Page 15:

Line 15-16: "Such profiles may indicate a non-steady-state fluid system." I think the authors should elaborate on this in the discussion.

Page 16:

Lines 4-8: "The porewater profile have complicated structures which can potentially be explained by…" The authors may want to consider adding a description of the complicated structures here and moving the interpretation to the discussion.

Page 17:

Fig.3: - NO4 should be NO3. - Consider adding DIC to the d13C label. - What are the black dots in the left lower panel? - Consider rescaling the NH4 axis. At present ammonium appears to be zero throughout the domain, while from Fig. 5 it's clear that it is not the case.

Page 23:

Line 14-15: "minerals, such as magnetitie, have been shown to dissolve when exposed to high concentration of sulfide in porewater for considerable time." This provides a strong argument to include sulfidic reduction of iron oxides in the model.

Page 26:

Fig. 5. Most fits look good. However, there appears to be a mismatch for the ammonium and phosphate profiles. This may indicate that the model does not resolve the remineralization of organic matter well. The authors should fix it or explain the problems in the text. Also there seems to be an issue with the $Mg^{2+}$ profiles.

Page 35:

Line 22-23: "We examined the entire reaction network. . . Fe(II) fluxes towards the oxic sediment layer" I find it slightly confusing that occasionally the authors seem to treat Fe(II) fluxes out of the model domain as fluxes towards the bottom-water and at other places as fluxes towards an oxic sediment layer not included in the model.

References:

Dale, A. W., Nickelsen, L., Scholz, F., Hensen, C., Oschlies, A., & Wallmann, K. (2015). A revised global estimate of dissolved iron fluxes from marine sediments. Global Biogeochemical Cycles, 29(5), 691-707.

Luff, R., & Wallmann, K. (2003). Fluid flow, methane fluxes, carbonate precipitation and biogeochemical turnover in gas hydrate-bearing sediments at Hydrate Ridge, Cascadia Margin: numerical modeling and mass balances. Geochimica et Cosmochimica Acta, 67(18), 3403-3421.

Luff, R., Wallmann, K., & Aloisi, G. (2004). Numerical modeling of carbonate crust formation at cold vent sites: significance for fluid and methane budgets and chemosynthetic biological communities. Earth and Planetary Science Letters, 221(1-4), 337-353.

Rickard, D. (2012). Sulfidic sediments and sedimentary rocks (Vol. 65). Newnes.

Rooze, J., Egger, M., Tsandev, I., & Slomp, C. P. (2016). Iron-dependent anaerobic oxidation of methane in coastal surface sediments: Potential controls and impact. Limnology and Oceanography, 61(S1).

Tryon, M. D., & Brown, K. M. (2004). Fluid and chemical cycling at Bush Hill: Implications for gas and hydrate rich environments. Geochemistry, Geophysics, Geosystems, 5(12).

Other references can be found in the manuscript.

---

## Referee Comment (RC2) · S. Kasten (Referee) · 4 Jul 2018

The contribution by Latour et al. investigates the biogeochemical cycling of Fe, S and C in cold-seep surface sediments around Svalbard and the continental shelf and fjords in northern Norway. A particular emphasis is put on the impact of the dynamic cycling and reaction pathways of these elements on the solute fluxes of $Fe^{2+}$ towards the upper oxic layer of these deposits and across the sediment/water interface. Rates of biogeochemical processes and fluxes of $Fe^{2+}$ towards the uppermost oxic surface sediments and into the bottom water were derived from transport/reaction modelling. Before I prepared my own evaluation report I had a look at the Interactive comment of Referee #1 and agree that several important (bio)geochemical reactions and transport processes (e.g. bioirrigation, mixing induced by bubble ebullition, etc.) that were previously shown

to be important at cold seep sites have not been considered. I also encourage the authors to consider these because some of these transport processes and reaction do significantly control the flux to and release of Fe from surface sediments. My major points are: Throughout the manuscript the authors speak of (direct) precipitation of pyrite. However, numerous studies have shown that pyrite formation, which can occur via several different pathways, does mostly not occur via direct precipitation from pore water but via several intermediate/precursor iron sulfide mineral phases. This should be considered throughout the manuscript and also be implemented into the reaction network/model. As a consequence what is schematically shown in Fig.2 – i.e. that HS- and Fe2+ directly react to form pyrite is not correct. Moreover, in most of the literature dealing with the transformation of precursor Fe sulfide minerals to pyrite it is discussed that the conversion into pyrite is primarily controlled by the availability of hydrogen sulfide. In this manuscript the authors only discuss Fe to be the limiting factor for further transformation to pyrite. I therefore recommend to also add a discussion about the significance of hydrogen sulfide and time (!) to exert a major control on transformation of iron sulfide precursor phases into pyrite and also cite the respective/relevant references. The authors have theoretically simulated the impact of changes in methane fluxes and input of Fe oxyhydroxides. What about the role of changes in organic matter fluxes and/or quality/reactivity over time? Changes in the burial flux of organic matter will certainly have a profound impact on the resulting rates of organoclastic iron and sulfate reduction in surface sediments and thus also determine the thickness of the Fe-rich zone and the steepness of the upward-directed dissolved Fe gradient – thus the flux of Fe2+ towards the uppermost oxic layer of the sediments and across the sediment/water interface - and the amount of iron sulfides formed (see specific comments below). I do not agree with the discussion about the usage of pyrite to trace former depth of the SMT both in the discussion chapter, the abstract and the conclusions. The authors question the use of pyrite – however their discussion and reference to relevant papers is not correct as it stands. Any statements and conclusions in this respect should thus be carefully re-considered and revised. With respect to the two

points raised above and the fact that the title of the manuscript is "dynamic interaction" I did somehow miss a reconstruction of the "real" geochemical and biogeochemical history of the sediments at the study sites - at least for a few sites representative of each of the three groups or for those where solid-phase data are available. The sensitivity tests presented remain a bit "hypothetical" for my taste and I would have liked to see a discussion of how the current geochemical zonation and the positions of reaction fronts fit to the distribution of the different Fe sulfide mineral phases determined at a few selected sites. If these do not fit – can this tell you something about past changes in methane fluxes or in the input flux of Fe oxihydroxides or amount and/or reactivity of organic carbon? Can you speculate on potential drivers, which have most likely caused these past variations at your study sites? I hope that the points given above and the specific comments listed below will help the authors to revise the manuscript. The English also needs a bit of polishing.

Specific comments

Page 2 l. 7: In marine sediments . . . . l. 16 ff.: . . . along global continental margins . . .; Has to be Niewöhner (please also correct this in the list of references); I suggest to add the following references here: Riedinger et al. (2005) and (2017)

Page 3 Ls. 21, 30: has to be "Wehrmann"

Page 4 l. 4: What exactly do you mean with "contrasting fluid seepage behavior"? This is not clear to me. Please specify. l. 10: So, what is the sedimentation rate? Would be interesting and of relevance to know. l. 18: . . . high methane concentration"s" – What are high methane concentrations? Please give the range of concentrations. Page 5 l. 14: What does the abbreviation "GHM3" stand for? Please explain. ls. 15-17: at the end of this sentence I propose to add: ". . . according to the procedure presented by Seeberg-Elverfeldt et al. (2005)". Page 6 Ls. 1-2: Was bottom water removed before the pore-water was collected by rhizons? Please explain.

Pages 6 and 7 In my version of the manuscript the title of the table was missing.

Page 9 l. 19: I guess you mean "gravity" cores instead of sediment cores (all the other samples you have worked on are also sediment cores) l. 32: content"s"

Page 10 Ls. 26/27: No, I do not agree with this statement. As already outlined above the reaction between Fe2+ and hydrogen sulfide does not produce pyrite directly but precursor iron sulfide minerals. Here, the authors state that they have excluded other intermediate sulfide minerals. However, numerous studies have shown that such intermediate iron sulfide sulfides or intermediate sulfur species precipitate at the Fe2+-sulfide reaction interface – as can also be seen from the study by Jørgensen et al. (2004) that they cite and other references (e.g., Kasten et al., 1998; Fu et al., 2008; review paper by Roberts in Earth-Science Reviews; Riedinger et al., 2017) and that the further conversion of such precursor phases to pyrite is often controlled by the availability of hydrogen sulfide. This is however not discussed at all in this manuscript. Page 10 ls. 31/32 and Page 13 l. 1: which kind of "dissolution" process precisely did you use in your model approach? Reductive dissolution?

Page 12 How did you determine the amount of iron oxide minerals in the surface sediments? At least the information and data are not given in the manuscript.

Page 13 l. 11: To my knowledge there are several papers by Verona Vandieken, Niko Finke and co-authors presenting hydrogen concentrations of marine pore waters.

Page 15 l. 17: There is a contradiction between the number of cores given here (12), in Table 1 (13) and Fig. caption 3. Fig. 3 A, B: What are the black dots in the left graphs (sulfate and HS) of several of the sites? Please explain this in the figure caption.

Ls. 18 ff. and throughout the manuscript: When referring to the shape of pore water profiles I prefer to speak of "steep" decrease/gradient rather that "rapid". l. 17/18: How do you know that sulfate turnover is "slow" . . . you only have pore water profiles which give you net rates. I would rather speak of a "minor decrease in sulfate concentrations with depth". The Results chapter already contains some interpretation. Please check this and potentially shift this to the Discussion chapter.

[Figure]

Page 23 l. 8 and throughout the manuscript: use "uppermost" instead of "first" ls. 15, 18, throughout the manuscript and in the references: has to be "März" ls. 18/19: Below "this" depth, the abundance of CRS "steeply" increases "coinciding" with a sharp decrease . . . .

Page 24 It would be good if the pore water profiles of Fe2+ and HS- would also be included/shown next to the graphs depicted in figure 4 because it would then be easier and straightforward to see where reaction fronts are currently located and how the position of these active reaction fronts compare to the depth distribution of the operationally defined iron sulfide minerals AVS and CRS. By the way, the susceptibility profiles of cores 904MC and 938MC look strange to me. Please check the data.

Page 25 l. 2 end: . . . in these fjord sediments

Page 31 l. 3: Who says that sulfate is the most abundant electron acceptor in marine environments? Please give the respective references – including key papers by B.B. Jørgensen on sulfate reduction in marine sediments. Perhaps also the paper by Bowles et al. (2014; Science) may be of interest for you in this context. l. 11: . . . suggests "a" tight coupling . . . ls. 22/23: Which interface precisely are you referring to here? Do you mean the diffusive interface where Fe2+ and HS react? Numerous papers – in particular those presented by Jørgensen and coworkers as well as Postma and Jakobsen (1996; GCA vol. 60) have shown that sulfate and iron reduction can co-occur within a broad depth interval or that the depth sequence in which Fe and sulfate reduction occur can even be reversed – depending on the reactivity of the available Fe (oxi)hydroxides.

Page 33 l. 2: Which microbial process precisely do you refer to here? Please give the relevant references. l. 27/28: . . . with very little "net" sulfate reduction . . . . As already mentioned above sulfate reduction can occur at considerable gross rates without any distinct decrease in pore-water sulfate concentrations as has been shown by numerous papers of Jørgensen and co-workers. l. 31: The fluxes of "which two compounds" are

you referring to here? Please explain.

Page 34 l.1 : Has to be "Wehrmann" l. 5: Which kind of "dynamics" precisely are you referring to here? This is not clear to me. l. 6/7: What is the role/contribution of the precipitation/formation of Fe oxihydroxides at the Fe redox boundary – i.e. at the upper boundary of the $Fe^{2+}$-rich zone? l. 10 ff.: As already mentioned above not only the upward flux of hydrogen sulfide from the zone of AOM but also co-occurrence of organoclastic iron and sulfate reduction can control the concentrations of $Fe^{2+}$ and the thickness/extent of the ferruginous zone. l. 11: I do not understand what precisely you mean with "initially" here? ... and also not in figure 12. l. 30 ff.: Pore-water Fe concentrations as those reported here have also been observed in other coastal marine depositional environments – also in settings not affected by active methane seepage (for example Oni et al. (2015; Frontiers in Microbiology).

Page 35 ls. 3 ff. until end of paragraph: I do not agree with the statement that "rapid formation of pyrite is generally observed at the SMT" and with the comments concerning the paper by Jørgensen et al. (2004). Formation of pyrite in relation to methane-mediated sulfate reduction occurs close to the SMT if the sulfidic zone is confined to a relatively thin zone around the SMT and if the SMT is fixed at a specific sediment depth for a prolonged period of time. This allows the initial precursor Fe sulfide mineral phases to be successively converted into pyrite by reaction with hydrogen sulfide continued to be produced at the SMT (e.g. Riedinger et al., 2005, 2017; März et al., 2008). Generally the formation of "AVS" (e.g. Fe monosulfides, greigite) occurs at the diffusional interface of $Fe^{2+}$ and hydrogen sulfide – generally also referred to as "sulfidization front" (e.g. Kasten et al., 1998, GCA; Riedinger et al., 2017). This reaction front can also nicely be seen in the data presented by Jørgensen et al. (2004) in Black Sea sediments, where AVS formation occurs at the current depth of the sulfidization front. Above this reaction front – i.e. in shallower sediments - abundant pyrite contents are found, which were formed during the downward migration of the SMT as a result of the deglacial flooding of the Black Sea with seawater and the profound increase in bottom water sulfate concentrations. During the downward migration of the SMT and the sulfidization front the uppermost sediments were permanently sulfidic and the initially formed AVS was further "matured" and converted into pyrite during ongoing exposure to/availability of hydrogen sulfide. You may also be interested in the paper by Henkel et al. (2012; GCA vol. 88) who modelled the downward migration of the SMT as a result of the deglacial flooding of the Black Sea with Mediterranean-sourced seawater. In this way – and in contrast to what the authors state here, in the abstract and in the conclusions - pyrite contents can thus well trace the past migration of the SMT and the sulfidization front. In addition, pyrite formation resulting from organoclastic sulfate reduction in surface sediments should also be discussed as a potential option to explain the occurrence of pyrite close to the sediment surface. To conclude, I suggest to discuss (and perhaps model) whether the finding of elevated pyrite contents close to the seafloor at sites 904MC and 1521GC may result from a downward migration of the SMT (and a thus a shallower depth position of the SMT and sulfidization front in the past) in response to a temporal decrease in the upward methane flux or from organoclastic sulfate reduction occurring close to the sediment surface.

Ls. 16 ff.: In chapter 5.3 the authors present and discuss the results of two sensitivity tests varying the fluxes of methane and the input of Fe oxides. Although the authors state on page 34 that organic matter is not the key driver of diagenetic/biogeochemical processes in the surface sediments at their study sites I suggest that changes in the input (both amount and quality/reactivity of TOC) may well impact the rates of biogeochemical processes and resulting solute fluxes – in particular of organoclastic sulfate reduction - within the surface sediments. It is not clear to me if and how they have varied organic matter burial fluxes in the two sensitivity tests. Please describe this more precisely. I agree that at the sites under profound impact of methane seepage (in particular the third group of sites) the temporal variation in the flux of TOC to the seafloor is certainly not a key driver. However, at the sites not dominated by upward methane fluxes, changes in the amount and/or reactivity/quality of organic matter will definitely have profound effects on rates of organoclastic Fe and sulfate reduction and resulting

diffusive fluxes – as well as the rate of iron sulfide formation (including pyrite) in the surface sediments.

L. 29: What do you mean with "disturbance" here?

Page 36 l. 2: Which "two mineral phases" precisely do you refer to here? Please explain. l. 9: increasing l. 14: does the word/term "scatterness" really exist?

Page 37 l. 2 – end of sentence: either: . . . is not a "variable" or . . . is not "variable" Caption of Fig. 10: . . . with variable methane supply . . ..

Page 38 l. 3: "milder" ? sounds odd l. 24: "this" flux? Please, say which flux precisely you mean here. l. 28: chemosynthesis-based

Page 39 l. 9: (oxihydr)oxide ls. 10, 12, 15, 16: I think it has to be "Oligobrachia" – please correct this accordingly here and in the caption of Fig. 11 - and check throughout the manuscript l. 16: What do you mean with "localized" sulfidic bottom water condition? l. 19: . . . complicated . . . that the bacteria that form these mats . . . l. 20: Please, precisely state which "speculation" you mean here.

Page 41 I find the schematic representation relatively hard to follow. You need to refer to or mention the boxes highlighted in red. I do not understand at all what precisely you mean with "initial increase" and "sequential increase" or "subsquential increase (please check spelling)? Please, describe this precisely here and in particular in the text.

Page 47 l. 17: Bohrmann, G. is missing as the last co-author of this paper. Please add.

Best regards, Sabine Kasten

————————————————

---

## Author Comment (AC1) · 6 Aug 2018

We appreciate the time and effort from this anonymous reviewer. We find the comments insightful and can substantially improve the quality of our work. The major criticisms are that there are a few unaccounted reactions/processes in our model, which may change the conclusions of this paper. We agree with most of these criticisms and will improve our model accordingly. However, we also want to keep the solution of our modeling unique, especially when it comes to the very complicated coupling between Fe and S. A certain degree of sacrifice and compromise is therefore necessary. For example, we are aware that aerobic processes are very important overall; however, due to the lack of any oxygen concentration measurements at our sites, modeling of these processes will be no more than educated speculation. Nonetheless, we are still

willing to implement these reactions to have an assessment on how critical these processes are. We reply to the major criticisms raised by the reviewer upfront and provide detailed point-to-point explanations later:

(1) Solute transport: The first criticism being that we consider only diffusion but not other solute transport mechanisms, such as bioturbation and/or water movement. We have ruled out the possibility of water movement (both upward and downward) in our earlier paper (Hong et al., 2017) by using one of the cores discussed in this paper (904MC). We will discuss this in more detail in the revised manuscript. As for bioturbation, we argue that, as a result of limited oxygen penetration for the majority of our sites, this is a less likely process to perturb the porewater system. The other support for our argument lies in our flux calculation (as presented in Table 1) where we did include the effect of bioturbation by assigning apparent "diffusion coefficients". These coefficients are two orders of magnitude smaller than the solute diffusion coefficients and therefore result in a very small contribution from bioturbation. We are aware that these "diffusion coefficients" for bioturbation are very crude estimations. Bioturbation may be important for the sites with rather bizarre structures in porewater profiles (e.g., 1029PC and 1123BLC). We will simulate the effect of bioturbation and discuss its importance for elemental cycles at these two sites in the revised manuscript.

(2) Pyrite formation: The second criticism that the reviewer has is about the pyrite formation pathway we assigned in the model. The assigned reaction pathway in our model is inspired by Rickard and Luther (2007), the same literature as the reviewer suggested. To better justify our choice, we summarize some of the conclusions from the paper. Rickard and Luther (2007) reviewed the processes proposed in the literatures about pyrite formation in section 7.4. The most often mentioned pathway: FeSm (the amorphous iron sulfide form) + S(0) → FeS2p (Berner et al.) was concluded by the authors to be "could not describe a mechanism since S(0) is in the form of S8, which would make this an impossible multimolecular reaction step". Rickard and Luther also concluded that "Certainly, it has been unequivocally demonstrated experimentally and

in natural systems that FeSm does not "transform to pyrite" in the sense of a solid-state transformation. FeSm, where it occurs, dissolves, and pyrite forms from the reaction between dissolved iron and sulfur species to which the products of the FeSm dissolution reaction contribute". This conclusion justifies our choice not to involve the amorphous iron sulfide phase in the reaction network. Rickard and Luther (2007) suggested that the Bunsen reaction (polysulfide controlled) and Berzelius reaction (sulfide controlled) are two more likely pathways, which involve an aqueous FeS intermediate (Eq, 45, 46, and 51 in the paper). More importantly, both pathways produce H2 as a by-product. Rickard and Luther (2007) also pointed out these two pathway can be fast under low-temperature condition with the presence of certain microbes operating sulfur disproportionation. As the formation of this aqueous intermediate, FeS, is not a rate-limiting step of the overall reaction, it is adequate to use the reaction we assigned in the model. In the revised version of the manuscript, we will also include newly-obtained d34S-CRS/AVS data to show that the two sulfur phases have undergone very different formation histories and may use different porewater sulfide pools.

(3) Aerobic processes: As shown by our porewater profiles, we argue that the sedimentary oxic layer is very thin (<1mm) for most of our sites. We will however implement a few aerobic processes, such as sulfide oxidation and nitrification, and assess how essential these reactions are with respect to the elemental cycles, despite the lack of oxygen measurement at any of our site to constrain these processes. We will emphasize that the modeling results can only be used to infer fluxes towards the oxic layer of sediments while the calculation results applying Fick's law are better for inferring fluxes leaving the sediment layer.

We will also amend the revised manuscript with supplementary tables showing the detailed reaction expressions, thermodynamic entries, diffusion coefficients, and initial/boundary conditions. Please see our point-to-point replies to other comments of the reviewer's from the attached file.

Please also note the supplement to this comment:
https://www.biogeosciences-discuss.net/bg-2018-223/bg-2018-223-AC1-
supplement.pdf

**Supplement:**

Reviewer #1:

We appreciate the time and effort from this anonymous reviewer. We find the comments insightful and can substantially improve the quality of our work. The major criticisms are that there are a few unaccounted reactions/processes in our model, which may change the conclusions of this paper. We agree with most of these criticisms and will improve our model accordingly. However, we also want to keep the solution of our modeling unique, especially when it comes to the very complicated coupling between Fe and S. A certain degree of sacrifice and compromise is therefore necessary. For example, we are aware that aerobic processes are very important overall; however, due to the lack of any oxygen concentration measurements at our sites, modeling of these processes will be no more than educated speculation. Nonetheless, we are still willing to implement these reactions to have an assessment on how critical these processes are. We reply to the major criticisms raised by the reviewer upfront and provide detailed point-to-point explanations later:

(1) **Solute transport:** The first criticism being that we consider only diffusion but not other solute transport mechanisms, such as bioturbation and/or water movement. We have ruled out the possibility of water movement (both upward and downward) in our earlier paper (Hong et al., 2017) by using one of the cores discussed in this paper (904MC). We will discuss this in more detail in the revised manuscript. As for bioturbation, we argue that, as a result of limited oxygen penetration for the majority of our sites, this is a less likely process to perturb the porewater system. The other support for our argument lies in our flux calculation (as presented in Table 1) where we did include the effect of bioturbation by assigning apparent "diffusion coefficients". These coefficients are two orders of magnitude smaller than the solute diffusion coefficients and therefore result in a very small contribution from bioturbation. We are aware that these "diffusion coefficients" for bioturbation are very crude estimations. Bioturbation may be important for the sites with rather bizarre structures in porewater profiles (e.g., 1029PC and 1123BLC). We will simulate the effect of bioturbation and discuss its importance for elemental cycles at these two sites in the revised manuscript.

(2) **Pyrite formation:** The second criticism that the reviewer has is about the pyrite formation pathway we assigned in the model. The assigned reaction pathway in our model is inspired by Rickard and Luther (2007), the same literature as the reviewer suggested. To better justify our choice, we summarize some of the conclusions from the paper. Rickard and Luther (2007) reviewed the processes proposed in the literatures about pyrite formation in section 7.4. The most often mentioned pathway: FeSm (the amorphous iron sulfide form) + S(0) → FeS2p (Berner et al.) was

concluded by the authors to be "*could not describe a mechanism since S(0) is in the form of S8, which would make this an impossible multimolecular reaction step*". Rickard and Luther also concluded that "*Certainly, it has been unequivocally demonstrated experimentally and in natural systems that FeSm does not "transform to pyrite" in the sense of a solid-state transformation. FeSm, where it occurs, dissolves, and pyrite forms from the reaction between dissolved iron and sulfur species to which the products of the FeSm dissolution reaction contribute*". This conclusion justifies our choice not to involve the amorphous iron sulfide phase in the reaction network. Rickard and Luther (2007) suggested that the Bunsen reaction (polysulfide controlled) and Berzelius reaction (sulfide controlled) are two more likely pathways, which involve an aqueous FeS intermediate (Eq, 45, 46, and 51 in the paper). More importantly, both pathways produce $H_2$ as a by-product. Rickard and Luther (2007) also pointed out these two pathway can be fast under low-temperature condition with the presence of certain microbes operating sulfur disproportionation. As the formation of this aqueous intermediate, FeS, is not a rate-limiting step of the overall reaction, it is adequate to use the reaction we assigned in the model. In the revised version of the manuscript, we will also include newly-obtained d34S-CRS/AVS data to show that the two sulfur phases have undergone very different formation histories and may use different porewater sulfide pools.

(3) **Aerobic processes:** As shown by our porewater profiles, we argue that the sedimentary oxic layer is very thin (<1mm) for most of our sites. We will however implement a few aerobic processes, such as sulfide oxidation and nitrification, and assess how essential these reactions are with respect to the elemental cycles, despite the lack of oxygen measurement at any of our site to constrain these processes. We will emphasize that the modeling results can only be used to infer fluxes towards the oxic layer of sediments while the calculation results applying Fick's law are better for inferring fluxes leaving the sediment layer.

We will also amend the revised manuscript with supplementary tables showing the detailed reaction expressions, thermodynamic entries, diffusion coefficients, and initial/boundary conditions. Please see our following point-to-point replies to other comments of the reviewer's.

The cycling of iron and sulfur in methane-rich sediments at various seep locations near Svalbard is investigated. Pore-water profiles are used to calibrate a reaction-transport model that takes into account intermediate organic matter breakdown species, such as acetate and molecular hydrogen (H2). This approach provides potentially interesting insights into the

coupling of various reaction pathways. For instance, they argue that pyrite formation may produce H2 and stimulate sulfate reduction, and that the accumulation of dissolved iron inhibits sulfate reduction. The calibrated model is then used to explore the effects of increasing iron oxide loading and methane influx on the turnover of chemical species and the partitioning between various reaction pathways. The data and model are also used to estimate dissolved iron fluxes to the overlying water. Although I find the modeling approach very interesting, a number of important reactions are missing. First of all, sulfidic reduction of iron oxides is not included, which is, as the authors acknowledge, important (page 23, line 14-15). For pyrite formation they use a reaction pathway in which H2 is produced. It may be more likely that most pyrite is formed through the FeS + S0 → FeS2 reaction, where FeS can be formed by sulfidic reduction of iron oxides and the Fe(II) + H2S reaction (see the specific comments).

From the low bottom-water temperature and the abundance of polychaetes (see paragraph 5.4) one may expect a high influx of oxygen into the sediment, but the model does not include oxygen and re-oxidation reactions.

*See our reply (3) above.*

This is likely to have a large impact on the estimated Fe(II) and H2 fluxes to the overlying water. In Fig. 5 there is a rapid decrease of ammonium in the upper 5 cm, which may be caused by nitrification or rapid transport/mixing.

*The reviewer may refer to the profiles from MC21 and MC26, where the ammonium concentration for the first 5 cm of the core is below ~10 microM. We agree that there may be unaccounted nitrification in the shallow sediments of these sites. We will implement this in the revised model.*

The data does not provide direct constraints on the transport in the model. The authors briefly mention methane seepage, gas bubble irrigation, and bioturbation as potential drivers of transport for the cores in Fig. 3C. In a cold seep one may expect up- or downward water flow (Tryon and Brown, 2004 and references therein), and the high number of large polychaetes makes it likely that also bioirrigation is important. All these flow processes could affect the profiles in Fig. 3A,B as well. The transport may have a large impact on the interpretation of pore profiles, so the authors may want to discuss these processes in more detail.

*Please see our (1) reply for this point. We will include the bioturbation simulation for cores 1029PC and 1123BLC.*

The results and discussion in the paper could also be better embedded within the context of other studies. Here are some examples that the authors may find interesting to consider: there are papers about reaction-transport models in cold seeps (e.g. Luff and Wallmann, 2003; Luff et al., 2004). There is a paper on carbon, iron, sulfide cycling in methane-rich sediments (Rooze et al., 2016), which may provide information about the importance of re-oxidation reactions and iron solids that are not included in the model. Rather critical I think are papers that discuss dissolved Fe fluxes to the overlying water (see paragraph 4.5 in Raiswell and Canfield, 2012; Dale et al. 2015). These papers stress the importance of aerobic iron re-oxidation and bioirrigation - factors that are ignored in the model.

*We appreciate the information from the reviewer. We will discuss our results in the context of these publications.*

In summary my main concerns are that a few important reactions are not included in the model and that the transport is not well constrained. Both factors could have a major influence on estimates of the dissolved iron fluxes. The reactions could be added to the model, and the authors may want to explore and describe the sensitivity of the model towards different transport regimes.

Specific comments:

Page 2: Line 8: "Redox reactions. . . carbon oxidized." To which 'order' does this sentence refer? If they refer to the stratification it is important to note that it is not always the case; for instance, bioirrigation, bubble irrigation, and transient diagenesis can mix the sequence up.

*Yes, we refer to the classical redox stratification. We will explain this in more details in the context of other processes as the reviewer pointed out.*

Line 18: "a quantitative. . . still lacking." This does not do justice to the body of literature that already exists and deals with cold seep biogeochemistry.

*We will change this statement.*

Line 20: "Previous studies. . ." Which studies exactly? The reduction of iron oxides also depends on other factors, such as the crystal structure and pH.

*We will revise this statement and explain the other factors.*

Line 24: Pyrite formation The authors focus on one mechanism of pyrite formation, which involves the production of H2. They do not include the FeS + S0 → FeS2 reaction, where the FeS can be formed by Fe + H2S + 2 HCO3 → FeS + 2 CO2 + 2 H2O and 2 FeOOH + 3 H2S

→ 2 FeS + S0 + 4 H2O. How likely is it that the reaction pathway in which H2 is produced occurs in Arctic sediments? Rickard and Luther (2007) briefly discuss this reaction (see reaction 29 in the article) and indicate that this reaction may only occur at higher temperatures. Can the authors make a case for the use of this particular reaction pathway and leaving out other pathways with FeS as an intermediate? Rickard (2012) provides a good overview of the various mechanisms that can produce pyrite in sediments.

*See our reply (2) for this comment.*

Line 30-32: "It is therefore expected. . . oxic layer in the sediment." Is this based on literature? Since dissolved Fe and sulfide react rapidly, would it not be more common that either all sulfide or all dissolved Fe is titrated out of the pore water?

*No, such statement is based on the consideration of the iron reduction and pyrite formation together. In situations of very high or low sulfide production rates, indeed either Fe(II) or sulfide will be completely titrated out of the porewater. However, as shown by most of our porewater profiles, we usually observed the transient condition of those two extreme cases. To some extent, we are interested in such transient condition and intent to investigate how the Fe and S cycles would be in such situation. We will modify this statement to better reflect our thoughts.*

Page 3: Line 7: "bioavailable" The authors make somewhat broad statements that should be backed up with references. Since Fe(III) can actually be taken up by microbes it is also bioavailable (Raiswell and Canfield, 2012).

*We will better define the term throughout the text.*

Line 8: "Soluble Fe(II) . . . can only be produced from anoxic environments" This is imprecise. Fe(II) can be produced in the water column, but most of it will be rapidly reoxidized.

*We will modify this statement.*

Page 5: Fig 1: The font size is too small.

*We will modify this.*

Page 10: Line 16: Perhaps it is better nomenclature to call these processes hydrogenotrophic and acetoclastic sulfate reduction.

*We will change the way we call these reaction.*

Line 30-31: "it is mathematically best to assume only the overall reaction" It is not clear to me how this can be proven mathematically. The explanation of the model is very terse. It is rather difficult to find all the information since the governing equations are not provided, there are no tables that sum up all the state variables, reaction stochiometries, and rate constants, and the information can only be found in other articles. The authors mention Hong et al. (2016) for which there is no entry in the bibliography. The other reference is to a supplementary file from a Nature Communications paper, but it seems only to give information about the reactions, not the transport.

*What we meant for that statement is, as we only have the concentrations of Fe(II) and hydrogen sulfide to constrain the production/consumption of both ions (which are already five reactions in our current model), we find it the best to minimize the set of reactions and only focus on the overall process. Unless there are additional data, such as isotopic signatures, it is likely that we have more variables than what can be constrained. We are sorry for such difficulties to the readers. We will provide supplementary tables with detailed rate expressions, stoichiometry information, and equilibrium constant calculations.*

Page 11: Fig 2: In the figure Co3 should be CO3. Sulfate is negatively charged. It takes effort to find all the labels in the figure, especially since so many different colors are used. For people with poor sight or bad printers it would be nice to have an overview of all the reactions in a table. It is not clear whether the reactions are reversible.

*We appreciate the suggestion. The figure will be amended and tables will be provided to explicitly state all reactions in the revised manuscript.*

Line 5-10: "For the reactions involving minerals (R1, R3, R9. . .." The paragraph starting here is a bit terse. Why did the authors choose this approach? The way this rate law is written implies that it is reversible, since $(1 - Q/K)$ can be positive and negative. To describe POC hydrolysis as a reversible equilibrium reaction appears troublesome to me.

*We apologize for the unclear statement about the reaction network. We will amend the paper with tables explicitly showing the reaction, boundary conditions, initial condition and other important factors used in the model. As for organic matter hydrolysis, this specific reaction is not reversible.*

Page 12: Line 4-5: "the equilibrium constants for other reactions were calculated. . . assuming 25 C" The actual temperature is ¬1 degree Celsius (table 1) and thus significantly lower than the 25 degree Celsius the authors use. It is a large difference. Why can the equilibrium constants not be corrected for the in-situ temperature?

*We will correct this.*

Line 9-10: "For reactions . . . Monod-type reaction with the basic form of" Why do the authors choose to use this rate expression? It's not clear whether the term (1-Q/Keq) act as an on/off switch or that the reaction is reversible.

*See our reply above. We will provide more detail information about the model setup. We can make the reactions irreversible in the model, such as for the case of POC hydrolysis.*

Line 19: "we set an imaginary mineral to produce methane. . ." Why do the authors choose this approach?

*This is to overcome the limitation from the software package we used, which does not allow us to define boundary conditions differently for different ions. In order to use "no-flux" lower boundary condition for most ions and "constant flux" lower boundary condition for methane, we figured the best way is to have an imaginary source of methane at the bottom of model regime.*

Line 23: "We used. . . as the initial condition" What are the initial conditions of the solids?

*See our reply above. We will provide detail information about this in the amended manuscript. All minerals, except for Fe(OH)3, were included in the initial condition. Fe(OH)3 is supplied from the top of sediment column by deposition.*

Line 27: "We then assigned the amount of iron hydroxide in the surface sediments (i.e. the upper boundary condition)" I guess that 'iron hydroxide' should be 'iron oxide'. It is not clear to me what is used as boundary condition for the solids. A table with all boundary conditions would be helpful.

*See our previous reply. This will be provided in the amended manuscript.*

Page 13: Lines 14-18: "For a. . . balance requirement" I am wondering how sensitive these fluxes are to computational errors. Would it not be better to calculate the fluxes across the top boundary directly?

*The fluxes across the top boundary condition (i.e., seafloor) have been estimated through the application of Fick's law. Such mass balance presented was meant for the consumption/production of materials within the model domain. We will present an additional sensitivity test to evaluate the influence of computational error.*

Line 26: There should be a minus sign in eq. 3.

*We have corrected this.*

Line 29-line 8(page 14): How did the authors calculate the specific values for the diffusion coefficients?

*These were calculated based on the temperature-dependent equations listed in Boudreau (1998). Seafloor temperatures at each site were used. The diffusion coefficients will be included in the new supplementary table.*

Line 29 suggests they were somehow corrected for different temperatures, but in line 32 it is stated that only one value was used. Why is the effect of bioturbation included for dissolved iron but not for sulfate? Middelburg's relationship for Db is a function of water depth, and is an average from different sites. However, given the rich seep biota it is likely that seeps form outliers with much higher Db values. The organization could perhaps be improved by making separate subsections about the reaction network, boundary conditions, and initial conditions. Information on the input of methane and other boundary conditions seems to be missing, and I also cannot find the parameterizations for the different fits. I don't recall reading anything about the advection of dissolved species.

*See our reply (1) for bioturbation. We will also consider bioturbation for sulfate in our flux calculation.*

Page 14: Equations 5 and 6: It is somewhat inconsistent that here the authors account for the re-oxidation of reduced iron, but not in the reaction-transport model.

*Similar to our reply (3) on aerobic processes, we see our modeling results only provide flux estimation to the oxic layer in the sediments (which is likely only < 1mm thick). On the other hand, the calculation from porewater profiles can be used to infer the bulk flux leaving the sediment layer (i.e., even considering the aerobic processes). We see the two estimations complimentary to each other. We will explain this in more details.*

Page 15: Line 15-16: "Such profiles may indicate a non-steady-state fluid system." I think the authors should elaborate on this in the discussion.

*We will elaborate on this and investigate these two sites with modeling including bioturbation.*

Page 16: Lines 4-8: "The porewater profile have complicated structures which can potentially be explained by. . ." The authors may want to consider adding a description of the complicated structures here and moving the interpretation to the discussion.

*Yes, we will have a new section elaborating on these two sites. See reply (1) above.*

Page 17: Fig.3: - NO4 should be NO3. - Consider adding DIC to the d13C label. - What are the black dots in the left lower panel? - Consider rescaling the NH4 axis. At present

ammonium appears to be zero throughout the domain, while from Fig. 5 it's clear that it is not the case.

*We will modify the figure accordingly.*

Page 23: Line 14-15: "minerals, such as magnetitie, have been shown to dissolve when exposed to high concentration of sulfide in porewater for considerable time." This provides a strong argument to include sulfidic reduction of iron oxides in the model.

*We agree that sulfidic reduction of iron oxide is important in the deeper sediments where sulfate is absent. However, as our main focus of the paper is on the processes in the iron and sulfate reduction zones, we decided not to implement this into our reaction network.*

Page 26: Fig. 5. Most fits look good. However, there appears to be a mismatch for the ammonium and phosphate profiles. This may indicate that the model does not resolve the remineralization of organic matter well. The authors should fix it or explain the problems in the text. Also there seems to be an issue with the $Mg(2+)$ profiles.

*We will modify the organic matter degradation to hopefully resolve the unsatisfactory fitting of ammonium and phosphate. As for Mg, we suspect some of the scattering of the data is due to analytical issues. We will double check this and explain it in the revised manuscript.*

Page 35: Line 22-23: "We examined the entire reaction network. . . Fe(II) fluxes towards the oxic sediment layer" I find it slightly confusing that occasionally the authors seem to treat Fe(II) fluxes out of the model domain as fluxes towards the bottom-water and at other places as fluxes towards an oxic sediment layer not included in the model.

*As we stated earlier in the reply, we meant to use the model results to infer fluxes to the sedimentary oxic layer and the calculation with Fick's law to infer the overall fluxes leaving the sediment layer. We will clarify this throughout the text.*

References: Dale, A. W., Nickelsen, L., Scholz, F., Hensen, C., Oschlies, A., & Wallmann, K. (2015). A revised global estimate of dissolved iron fluxes from marine sediments. Global Biogeochemical Cycles, 29(5), 691-707. Luff, R., & Wallmann, K. (2003). Fluid flow, methane fluxes, carbonate precipitation and biogeochemical turnover in gas hydrate-bearing sediments at Hydrate Ridge, Cascadia Margin: numerical modeling and mass balances. Geochimica et Cosmochimica Acta, 67(18), 3403-3421. Luff, R., Wallmann, K., & Aloisi, G. (2004). Numerical modeling of carbonate crust formation at cold vent sites: significance for fluid and methane budgets and chemosynthetic biological communities. Earth and Planetary Science Letters, 221(1-4), 337-353. Rickard, D. (2012). Sulfidic sediments and sedimentary rocks (Vol. 65). Newnes. Rooze, J., Egger, M., Tsandev, I., & Slomp, C. P. (2016). Irondependent anaerobic oxidation of methane in coastal surface sediments: Potential controls and impact. Limnology and Oceanography, 61(S1). Tryon, M. D., & Brown, K. M. (2004). Fluid and chemical cycling at Bush Hill: Implications for gas and hydrate rich environments. Geochemistry, Geophysics, Geosystems, 5(12). Other references can be found in the manuscript.

---

## Author Comment (AC2) · 6 Aug 2018

We appreciate the comments and feedbacks from Prof. Dr. Sabine Kasten, which we find them very helpful. We agree with most of the comments from Dr. Kasten and will gladly revise the manuscript accordingly. Similar to what we responded to reviewer #1, we would like to point out that one of the greatest challenges is to have enough constraints for the model results, especially in the case of pyrite formation. We are aware that our current model has only a simplified reaction pathway. However, this is mostly due to the limited constraints we have; i.e., we have only porewater sulfide and Fe(II) concentrations and CRS abundance to constrain three different processes in the current model. We feel like this is perhaps the optimal setup we can have at the moment. Future improvement is possible with constraints from the isotopic signatures

of these porewater species and information from other intermediate sulfur species. Here we reply to the major concerns of the reviewer upfront and provide detailed point-to-point explanations later.

(1) The formation pathway of pyrite: We reply to the same question from reviewer #1 as follow. The assigned reaction pathway in our model is inspired by Rickard and Luther (2007), the same literature as the reviewer #1 suggested. To better justify our choice, we summarize some of the conclusions from the paper. Rickard and Luther (2007) reviewed the processes proposed in the literatures about pyrite formation in section 7.4. The most often mentioned pathway: FeSm (the amorphous iron sulfide form) + S(0) → FeS2p (Berner et al.) was concluded by the authors to be "could not describe a mechanism since S(0) is in the form of S8, which would make this an impossible multi-molecular reaction step". Rickard and Luther also concluded that "Certainly, it has been unequivocally demonstrated experimentally and in natural systems that FeSm does not "transform to pyrite" in the sense of a solid-state transformation. FeSm, where it occurs, dissolves, and pyrite forms from the reaction between dissolved iron and sulfur species to which the products of the FeSm dissolution reaction contribute". This conclusion justifies our choice not to involve the amorphous iron sulfide phase in the reaction network. Rickard and Luther (2007) suggested that the Bunsen reaction (polysulfide controlled) and Berzelius reaction (sulfide controlled) are two more likely pathways, in which these reactions involve an aqueous FeS intermediate (Eq, 45, 46, and 51 in the paper). More importantly, both pathways produce H2 as a by-product. Rickard and Luther (2007) also pointed out these two pathway can be fast under low-temperature condition with the presence of certain microbes operating sulfur disproportionation. As the formation of this aqueous intermediate, FeS, is not a rate-limiting step of the overall reaction, it is adequate to use the reaction we assigned in the model. We also include some newly-obtained d34S data from both CRS and AVS to shown that the two sulfur fractions underwent very different history and may use different pools of hydrogen sulfide when forming. We feel that these new data included will also help justify our choice of pyrite formation pathways.

(2) The contribution from organic matter: We entirely agree with the reviewer's opinion that the deposition of organic matter will also have large impact on the Fe cycle. In the original manuscript, we discussed briefly this affect by comparing two sites with different organic matter degradation pathways (Fig. 9). We are aware that this may not be obvious for the readers so we will extend this discussion and focus a bit more on this aspect as the reviewer requested.

(3) The application of pyrite abundance as paleo-SMT indicator: Thanks for the suggestions from Dr. Kasten, we will revise the literatures we cited in the discussion of sulfur record in the Black Sea. We will also soften our criticism about the application of pyrite as paleo-SMT indicator. We agree such application may work in other locations. However, based on the data we presented for some of our sites, we see that pyrite abundance is not necessary applicable for this purpose at least for our study sites.

Please see our point-to-point reply in the attached file.

Please also note the supplement to this comment:
https://www.biogeosciences-discuss.net/bg-2018-223/bg-2018-223-AC2-supplement.pdf

**Supplement:**

Reviewer #2 (Prof. Dr. Sabine Kasten)

We appreciate the comments and feedbacks from Prof. Dr. Sabine Kasten, which we find them very helpful. We agree with most of the comments from Dr. Kasten and will gladly revise the manuscript accordingly. Similar to what we responded to reviewer #1, we would like to point out that one of the greatest challenges is to have enough constraints for the model results, especially in the case of pyrite formation. We are aware that our current model has only a simplified reaction pathway. However, this is mostly due to the limited constraints we have; i.e., we have only porewater sulfide and Fe(II) concentrations and CRS abundance to constrain three different processes in the current model. We feel like this is perhaps the optimal setup we can have at the moment. Future improvement is possible with constraints from the isotopic signatures of these porewater species and information from other intermediate sulfur species. Here we reply to the major concerns of the reviewer upfront and provide detailed point-to-point explanations later.

(1) **The formation pathway of pyrite:** We reply to the same question from reviewer #1 as follow. The assigned reaction pathway in our model is inspired by Rickard and Luther (2007), the same literature as the reviewer #1 suggested. To better justify our choice, we summarize some of the conclusions from the paper. Rickard and Luther (2007) reviewed the processes proposed in the literatures about pyrite formation in section 7.4. The most often mentioned pathway: FeSm (the amorphous iron sulfide form) + S(0) → FeS2p (Berner et al.) was concluded by the authors to be "*could not describe a mechanism since S(0) is in the form of S8, which would make this an impossible multimolecular reaction step*". Rickard and Luther also concluded that "*Certainly, it has been unequivocally demonstrated experimentally and in natural systems that FeSm does not "transform to pyrite" in the sense of a solid-state transformation. FeSm, where it occurs, dissolves, and pyrite forms from the reaction between dissolved iron and sulfur species to which the products of the FeSm dissolution reaction contribute*". This conclusion justifies our choice not to involve the amorphous iron sulfide phase in the reaction network. Rickard and Luther (2007) suggested that the Bunsen reaction (polysulfide controlled) and Berzelius reaction (sulfide controlled) are two more likely pathways, in which these reactions involve an aqueous FeS intermediate (Eq, 45, 46, and 51 in the paper). More importantly, both pathways produce $H_2$ as a by-product. Rickard and Luther (2007) also pointed out these two pathway can be fast under low-temperature condition with the presence of certain microbes operating sulfur disproportionation. As the formation of this aqueous intermediate, FeS, is not a rate-limiting step of the overall reaction, it is adequate to use the reaction we assigned in the model.

     We also include some newly-obtained d34S data from both CRS and AVS to shown that the two sulfur fractions underwent very different history and may use different pools of hydrogen sulfide when forming. We feel that these new data included will also help justify our choice of pyrite formation pathways.

(2) **The contribution from organic matter:** We entirely agree with the reviewer's opinion that the deposition of organic matter will also have large impact on the Fe cycle. In the original manuscript, we discussed briefly this affect by comparing two sites with different organic matter degradation pathways (Fig. 9). We are aware that this may not be obvious for the readers so we will extend this discussion and focus a bit more on this aspect as the reviewer requested.

(3) **The application of pyrite abundance as paleo-SMT indicator:** Thanks for the suggestions from Dr. Kasten, we will revise the literatures we cited in the discussion of sulfur record in the Black Sea. We will also soften our criticism about the application of pyrite as paleo-SMT indicator. We agree such application may work in other locations. However, based on the data we presented for some of our sites, we see that pyrite abundance is not necessary applicable for this purpose at least for our study sites.

The contribution by Latour et al. investigates the biogeochemical cycling of Fe, S and C in cold-seep surface sediments around Svalbard and the continental shelf and fjords in northern Norway. A particular emphasis is put on the impact of the dynamic cycling and reaction pathways of these elements on the solute fluxes of $Fe^{2+}$ towards the upper oxic layer of these deposits and across the sediment/water interface. Rates of biogeochemical processes and fluxes of $Fe^{2+}$ towards the uppermost oxic surface sediments and into the bottom water were derived from transport/reaction modelling. Before I prepared my own evaluation report I had a look at the Interactive comment of Referee #1 and agree that several important (bio)geochemical reactions and transport processes (e.g. bioirrigation, mixing induced by bubble ebullition, etc.) that were previously shown to be important at cold seep sites have not been considered. I also encourage the authors to consider these because some of these transport processes and reaction do significantly control the flux to and release of Fe from surface sediments.

My major points are: Throughout the manuscript the authors speak of (direct) precipitation of pyrite. However, numerous studies have shown that pyrite formation, which can occur via several different pathways, does mostly not occur via direct precipitation from pore water but via several intermediate/precursor iron sulfide mineral phases. This should be considered throughout the manuscript and also be implemented into the reaction network/model. As a

consequence what is schematically shown in Fig.2 – i.e. that HS and Fe2+ directly react to form pyrite is not correct.

*Please see our reply (1) above.*

Moreover, in most of the literature dealing with the transformation of precursor Fe sulfide minerals to pyrite it is discussed that the conversion into pyrite is primarily controlled by the availability of hydrogen sulfide. In this manuscript the authors only discuss Fe to be the limiting factor for further transformation to pyrite. I therefore recommend to also add a discussion about the significance of hydrogen sulfide and time (!) to exert a major control on transformation of iron sulfide precursor phases into pyrite and also cite the respective/relevant references.

*We agree that sulfide plays an important role in the formation of pyrite and we are willing to discuss this factor (together with time) in the revised manuscript. We also would like to stress that our intension for the current work is to emphasize more the significance of Fe(II) supply in pyrite formation, especially in the topmost 10 cm of sediments. Our solid phase data clearly show a rapid increase in CRS abundance at the base of the iron reduction zone, where porewater Fe(II) is almost undetectable and porewater sulfide concentration starts to rapidly increase. We feel like this is a less discussed aspect when it comes to pyrite formation and we would like to have a bit more emphasis on this process.*

The authors have theoretically simulated the impact of changes in methane fluxes and input of Fe oxyhydroxides. What about the role of changes in organic matter fluxes and/or quality/reactivity over time? Changes in the burial flux of organic matter will certainly have a profound impact on the resulting rates of organoclastic iron and sulfate reduction in surface sediments and thus also determine the thickness of the Fe-rich zone and the steepness of the upward-directed dissolved Fe gradient – thus the flux of Fe2+ towards the uppermost oxic layer of the sediments and across the sediment/water interface - and the amount of iron sulfides formed (see specific comments below).

*We agree that deposition of organic matter will also have a great impact on benthic Fe(II) fluxes. We briefly discussed this factor by having the sensitivity test on sites with different organic matter abundance and sulfate reduction pathways (organoclastic vs. AOM-coupled) as shown in Fig. 9. We agree this may not be that obvious for the readers and we will expand this discussion a bit and/or amended with some more sensitivity test to explicitly discuss the influence of organic matter deposition.*

I do not agree with the discussion about the usage of pyrite to trace former depth of the SMT both in the discussion chapter, the abstract and the conclusions. The authors question the use of pyrite – however their discussion and reference to relevant papers is not correct as it stands. Any statements and conclusions in this respect should thus be carefully re-considered and revised.

*We appreciate the comments and we will revise the text and references to tone-down a bit in this section. It is not our intension to accuse these earlier works using pyrite abundance as a paleo-indicator of SMT. However, based on the solid phase data from two of our cores and the porewater data from all the other cores, we intend to show that significant pyrite formation can occur at the interface between the iron reduction zone and sulfate reduction zone; such formation of pyrite is not directly linked to methane flux (or the depth of SMT) but is also controlled by iron supply. At least from the one core (1521GC) we have data showing pyrite formation is much more prominent in the surficial sediments than at the depth of SMT.*

With respect to the two points raised above and the fact that the title of the manuscript is "dynamic interaction" I did somehow miss a reconstruction of the "real" geochemical and biogeochemical history of the sediments at the study sites - at least for a few sites representative of each of the three groups or for those where solid-phase data are available. The sensitivity tests presented remain a bit "hypothetical" for my taste and I would have liked to see a discussion of how the current geochemical zonation and the positions of reaction fronts fit to the distribution of the different Fe sulfide mineral phases determined at a few selected sites. If these do not fit – can this tell you something about past changes in methane fluxes or in the input flux of Fe oxihydroxides or amount and/or reactivity of organic carbon? Can you speculate on potential drivers, which have most likely caused these past variations at your study sites?

*We agree that there was not too much discussion of reaction zones on the solid phase data. We will have a section discussing the reaction zones through the comparison of porewater and solid phase geochemistry (including the new d34S-CRS/AVS we obtained) as the reviewer suggested.*

I hope that the points given above and the specific comments listed below will help the authors to revise the manuscript. The English also needs a bit of polishing.

*We appreciate all the comments and suggestions from the reviewer. We will polish the language to meet the expectation.*

Specific comments:

Page 2 l. 7: In marine sediments . . .. l. 16 ff.: . . . along global continental margins . . .; Has to be Niewöhner (please also correct this in the list of references); I suggest to add the following references here: Riedinger et al. (2005) and (2017)

*We will include these references.*

Page 3 Ls. 21, 30: has to be "Wehrmann"

*Thanks for the correction.*

Page 4 l. 4: What exactly do you mean with "contrasting fluid seepage behavior"? This is not clear to me. Please specify.

*This sentence will be modified to "Seepage of methane in different phases, gaseous vs. dissolved, were observed...."*

l. 10: So, what is the sedimentation rate? Would be interesting and of relevance to know.

*We have added this information. The sedimentation estimated from one gravity core is 9 cm/kyr for the first meter of sediment and 80 cm/kyr from 1-3 mbsf.*

l. 18: . . . high methane concentration"s" – What are high methane concentrations? Please give the range of concentrations.

*We have added the range of concentration (20-60 nM).*

Page 5 l. 14: What does the abbreviation "GHM3" stand for? Please explain.

*We have added the explanation.*

ls. 15-17: at the end of this sentence I propose to add: ". . . according to the procedure presented by Seeberg-Elverfeldt et al. (2005)".

*We have amended the sentence.*

Page 6 Ls. 1-2: Was bottom water removed before the pore-water was collected by rhizons? Please explain.

*We always kept ~5-10 cm of coretop water in the linear while Rhizons were collecting the porewater to prevent oxygen invasion in the sediments.*

Pages 6 and 7 In my version of the manuscript the title of the table was missing.

*We will check the table.*

Page 9 l. 19: I guess you mean "gravity" cores instead of sediment cores (all the other samples you have worked on are also sediment cores)

*Only 1521GC is a gravity core; the other two are multi cores.*

l. 32: content"s"

*We have corrected this.*

Page 10 Ls. 26/27: No, I do not agree with this statement. As already outlined above the reaction between Fe2+ and hydrogen sulfide does not produce pyrite directly but precursor iron sulfide minerals. Here, the authors state that they have excluded other intermediate sulfide minerals. However, numerous studies have shown that such intermediate iron sulfide sulfides or intermediate sulfur species precipitate at the Fe2+- sulfide reaction interface – as can also be seen from the study by Jørgensen et al. (2004) that they cite and other references (e.g., Kasten et al., 1998; Fu et al., 2008; review paper by Roberts in Earth-Science Reviews; Riedinger et al., 2017) and that the further conversion of such precursor phases to pyrite is often controlled by the availability of hydrogen sulfide. This is however not discussed at all in this manuscript.

*Please see our reply (1) above.*

Page 10 ls. 31/32 and Page 13 l. 1: which kind of "dissolution" process precisely did you use in your model approach? Reductive dissolution?

*Such dissolution is only controlled by the concentration of Fe(II) in the porewater; i.e., dissolution occurs when Fe(II) concentration drops to a certain level.*

Page 12 How did you determine the amount of iron oxide minerals in the surface sediments? At least the information and data are not given in the manuscript.

*This is only obtained by data fitting. We were able to constrain how much Fe is fixed in pyrite minerals and how fast Fe(II) is produced (from porewater Fe(II) profile). The assigned abundance of iron oxide should be sufficient to satisfy these two constraints. We will explain this better in the text.*

Page 13 l. 11: To my knowledge there are several papers by Verona Vandieken, Niko Finke and co-authors presenting hydrogen concentrations of marine pore waters.

*Thanks for the information. We will add these references to the list.*

Page 15 l. 17: There is a contradiction between the number of cores given here (12), in Table 1 (13) and Fig. caption 3.

*The different numbers of cores listed in the text and table/figure are correct. We only categorize the 12 multi-cores while 13 cores (including the gravity cores) were reported in the paper. We will try to clarify this statement a bit.*

Fig. 3 A, B: What are the black dots in the left graphs (sulfate and HS) of several of the sites? Please explain this in the figure caption.
*We will correct this.*

Ls. 18 ff. and throughout the manuscript: When referring to the shape of pore water profiles I prefer to speak of "steep" decrease/gradient rather that "rapid".
*We will correct this.*

l. 17/18: How do you know that sulfate turnover is "slow" . . . you only have pore water profiles which give you net rates. I would rather speak of a "minor decrease in sulfate concentrations with depth". The Results chapter already contains some interpretation. Please check this and potentially shift this to the
*We will modify the statement. It is our intention to provide some preliminary discussions in the result section when describing the profiles, so that the discussion part can be more dedicated to the other aspects that require more attention.*

Page 23 l. 8 and throughout the manuscript: use "uppermost" instead of "first" ls.
*We will change this throughout the text.*

15, 18, throughout the manuscript and in the references: has to be "März" ls.
*We will correct this.*
18/19: Below "this" depth, the abundance of CRS "steeply" increases "coinciding" with a sharp decrease . . ..
*This will be modified.*

Page 24 It would be good if the pore water profiles of Fe2+ and HS- would also be included/shown next to the graphs depicted in figure 4 because it would then be easier and straightforward to see where reaction fronts are currently located and how the position of these active reaction fronts compare to the depth distribution of the operationally defined iron sulfide minerals AVS and CRS. By the way, the susceptibility profiles of cores 904MC and 938MC look strange to me. Please check the data.

*Yes, we will modify this figure and place porewater data alongside the solid phase data and have discussion about the reaction zones. We will check the MagSus data from those two cores.*

Page 25 l. 2 end: . . . in these fjord sediments

*We have modified this.*

Page 31 l. 3: Who says that sulfate is the most abundant electron acceptor in marine environments? Please give the respective references – including key papers by B.B. Jørgensen on sulfate reduction in marine sediments. Perhaps also the paper by Bowles et al. (2014; Science) may be of interest for you in this context.

l. 11: . . . suggests "a" tight coupling . . . ls. 22/23: Which interface precisely are you referring to here? Do you mean the diffusive interface where Fe2+ and HS react? Numerous papers – in particular those presented by Jørgensen and coworkers as well as Postma and Jakobsen (1996; GCA vol. 60) have shown that sulfate and iron reduction can cooccur within a broad depth interval or that the depth sequence in which Fe and sulfate reduction occur can even be reversed – depending on the reactivity of the available Fe (oxi)hydroxides.
*We will include these references for the statements.*

Page 33 l. 2: Which microbial process precisely do you refer to here? Please give the relevant references.

l. 27/28: . . . with very little "net" sulfate reduction . . .. As already mentioned above sulfate reduction can occur at considerable gross rates without any distinct decrease in pore-water sulfate concentrations as has been shown by numerous papers of Jørgensen and co-workers.
*We will modify the statement in the context of references suggested.*

l. 31: The fluxes of "which two compounds" are you referring to here? Please explain.
*We will explain this.*

Page 34 l.1 : Has to be "Wehrmann"
*We will check and correct this throughout the text.*

l. 5: Which kind of "dynamics" precisely are you referring to here? This is not clear to me.
*We will explain this.*

l. 6/7: What is the role/contribution of the precipitation/formation of Fe oxihydroxides at the Fe redox boundary – i.e. at the upper boundary of the Fe2+-rich zone?

*This was not accounted for in the model. The abundance of Fe oxyhydroxides is assigned as the top boundary condition, which can be from the water column or from the precipitation as the reviewer suggested. We will explain this in the revised manuscript.*

l. 10 ff.: As already mentioned above not only the upward flux of hydrogen sulfide from the zone of AOM but also co-occurrence of organoclastic iron and sulfate reduction can control the concentrations of Fe2+ and the thickness/extent of the ferruginous zone.

*This is true and we have intended to show this in fig. 9. As our reply earlier, we will expand this part and discuss the role of organic matter in the Fe cycle.*

l. 11: I do not understand what precisely you mean with "initially" here? . . . and also not in figure 12.

*We will explain this better.*

l. 30 ff.: Pore-water Fe concentrations as those reported here have also been observed in other coastal marine depositional environments – also in settings not affected by active methane seepage (for example Oni et al. (2015; Frontiers in Microbiology).

*We will include the reference suggested.*

Page 35 ls. 3 ff. until end of paragraph: I do not agree with the statement that "rapid formation of pyrite is generally observed at the SMT" and with the comments concerning the paper by Jørgensen et al. (2004). Formation of pyrite in relation to methane-mediated sulfate reduction occurs close to the SMT if the sulfidic zone is confined to a relatively thin zone around the SMT and if the SMT is fixed at a specific sediment depth for a prolonged period of time. This allows the initial precursor Fe sulfide mineral phases to be successively converted into pyrite by reaction with hydrogen sulfide continued to be produced at the SMT (e.g. Riedinger et al., 2005, 2017; März et al., 2008). Generally the formation of "AVS" (e.g. Fe monosulfides, greigite) occurs at the diffusional interface of Fe2+ and hydrogen sulfide – generally also referred to as "sulfidization front" (e.g. Kasten et al., 1998, GCA; Riedinger et al., 2017). This reaction front can also nicely be seen in the data presented by Jørgensen et al. (2004) in Black Sea sediments, where AVS formation occurs at the current depth of the sulfidization front. Above this reaction front – i.e. in shallower sediments - abundant pyrite contents are found, which were formed during the downward migration of the SMT as a result of the deglacial flooding of the Black Sea with seawater and the profound increase in bottom water sulfate concentrations. During the downward migration of the SMT and the sulfidization front

the uppermost sediments were permanently sulfidic and the initially formed AVS was further "matured" and converted into pyrite during ongoing exposure to/availability of hydrogen sulfide. You may also be interested in the paper by Henkel et al. (2012; GCA vol. 88) who modelled the downward migration of the SMT as a result of the deglacial flooding of the Black Sea with Mediterranean-sourced seawater. In this way – and in contrast to what the authors state here, in the abstract and in the conclusions - pyrite contents can thus well trace the past migration of the SMT and the sulfidization front. In addition, pyrite formation resulting from organoclastic sulfate reduction in surface sediments should also be discussed as a potential option to explain the occurrence of pyrite close to the sediment surface. To conclude, I suggest to discuss (and perhaps model) whether the finding of elevated pyrite contents close to the seafloor at sites 904MC and 1521GC may result from a downward migration of the SMT (and a thus a shallower depth position of the SMT and sulfidization front in the past) in response to a temporal decrease in the upward methane flux or from organoclastic sulfate reduction occurring close to the sediment surface.

*Please see our reply (3) above.*

Ls. 16 ff.: In chapter 5.3 the authors present and discuss the results of two sensitivity tests varying the fluxes of methane and the input of Fe oxides. Although the authors state on page 34 that organic matter is not the key driver of diagenetic/biogeochemical processes in the surface sediments at their study sites I suggest that changes in the input (both amount and quality/reactivity of TOC) may well impact the rates of biogeochemical processes and resulting solute fluxes – in particular of organoclastic sulfate reduction - within the surface sediments. It is not clear to me if and how they have varied organic matter burial fluxes in the two sensitivity tests. Please describe this more precisely. I agree that at the sites under profound impact of methane seepage (in particular the third group of sites) the temporal variation in the flux of TOC to the seafloor is certainly not a key driver. However, at the sites not dominated by upward methane fluxes, changes in the amount and/or reactivity/quality of organic matter will definitely have profound effects on rates of organoclastic Fe and sulfate reduction and resulting diffusive fluxes – as well as the rate of iron sulfide formation (including pyrite) in the surface sediments.

*We appreciate the suggestion from the reviewer. We entirely agree that organic matter can have a big impact on Fe cycling, even though it may not be that significant at our sites. We have briefly examined this by comparing the two sites with different organic matter content (as shown in fig9 (a&B) vs. (c&d)). We now realized our intention was not obvious to the reader. We will extent this part and have a more thorough discussion on the effect of organic matter.*

L. 29: What do you mean with "disturbance" here?

*We meant the different amounts of iron (oxyhydr)oxides from water column. We will modify this.*

Page 36 l. 2: Which "two mineral phases" precisely do you refer to here? Please explain.

*We meant between (oxyhydr)oxides and other iron-containing minerals such as goethite. We will clarify this.*

l. 9: increasing

*corrected.*

l. 14: does the word/term "scatterness" really exist?

*It has been changed to scattering.*

Page 37 l. 2 – end of sentence: either: . . . is not a "variable" or . . . is not "variable" Caption of Fig. 10: . . . with variable methane supply . . ..

*We have changed the sentence to "seafloor iron (oxyhydr)oxide input is constant".*

Page 38 l. 3: "milder" ? sounds odd l. 24: "this" flux? Please, say which flux precisely you mean here. l. 28: chemosynthesis-based

*We have corrected these.*

Page 39 l. 9: (oxihydr)oxide ls. 10, 12, 15, 16: I think it has to be "Oligobrachia" – please correct this accordingly here and in the caption of Fig. 11 - and check throughout the manuscript

*The spelling of (oxyhydr)oxide is correct and the same as in other literatures. We have corrected this throughout the text and caption.*

l. 16: What do you mean with "localized" sulfidic bottom water condition?

*We meant the local bottom water may be rich in sulfide concentration. We will clarify this.*

l. 19: . . . complicated . . . that the bacteria that form these mats . . . l. 20: Please, precisely state which "speculation" you mean here.

*We meant the speculation that the bottom water sulfide concentration is higher in the region where Fig 11C was taken. We will clarify this statement.*

Page 41 I find the schematic representation relatively hard to follow. You need to refer to or mention the boxes highlighted in red. I do not understand at all what precisely you mean with

"initial increase" and "sequential increase" or "subsquential increase (please check spelling)?
Please, describe this precisely here and in particular in the text.

*We will clarify this and find another way to illustrate what we meant in Fig. 12.*

Page 47 l. 17: Bohrmann, G. is missing as the last co-author of this paper.

*We have corrected this.*

Please add. Best regards, Sabine Kasten